# Adaptive Cannistraci-Hebb Network Automata Modelling of Complex Networks for Path-based Link Prediction

Jialin Zhao[1,2]     Alessandro Muscoloni[1,3,4]     Umberto Michieli[5,6]     Yingtao Zhang[1,2]

Carlo Vittorio Cannistraci[1,2,3,4] *

[1]Center for Complex Network Intelligence (CCNI) [†]
[2]Dept. of Computer Science & Technology, [3]School of Biomedical Engineering, Tsinghua University
[4]Biomedical Cybernetics Group, Technische Universität Dresden, Germany [‡]
[5]University of Padova, Italy, [6]Canva Research

## Abstract

Many complex networks have partially observed or evolving connectivity, making link prediction a fundamental task. Topological link prediction infers missing links using only network topology, with applications in social, biological, and technological systems. The Cannistraci-Hebb (CH) theory provides a topological formulation of Hebbian learning, grounded on two pillars: (1) the **minimization of external links** within local communities, and (2) the **path-based definition of local communities** that capture homophilic (similarity-driven) interactions via paths of length 2 and synergetic (diversity-driven) interactions via paths of length 3. Building on this, we introduce the Cannistraci-Hebb Adaptive (CHA) network automata, an adaptive learning machine that automatically selects the optimal CH rule and path length to model each network. CHA unifies theoretical interpretability and data-driven adaptivity, bridging physics-inspired network science and machine intelligence. Across 1,269 networks from 14 domains, CHA consistently surpasses state-of-the-art methods—including SPM, SBM, graph embedding methods, and message-passing graph neural networks—while revealing the mechanistic principles governing link formation. Our code is available at `https://github.com/biomedical-cybernetics/Cannistraci_Hebb_network_automata`.

## 1 Introduction

Many complex networks have a connectivity that might be only partially detected or that tends to grow over time, hence the prediction of non-observed links is a fundamental problem in network science. The aim of topological link prediction is to forecast these non-observed links by only exploiting features intrinsic to the network topology. It has a wide range of real applications, like suggesting friendships in social networks or predicting interactions in biological networks [1–3]. A plethora of methods based on different methodological principles have been developed in recent years, and in this

---

[*]Corresponding author, kalokagathos.agon@gmail.com

[†]Research center in Tsinghua Laboratory of Brain and Intelligence (THBI), Department of Psychological and Cognitive Sciences

[‡]Past affiliation of the research group in Biotechnology Center (BIOTEC), Center for Molecular and Cellular Bioengineering (CMCB), Center for Systems Biology Dresden (CSBD), Cluster of Excellence Physics of Life (PoL), Department of Physics, Technische Universität Dresden, Tatzberg 47/49, 01307 Dresden, Germany

39th Conference on Neural Information Processing Systems (NeurIPS 2025).

study we will consider for reference the state-of-the-art algorithms. The first is Structural Perturbation Method (SPM), a model-free global approach that relies on a theory derived from the first-order perturbation in quantum mechanics [4]. The second represents a class of generative models named Stochastic Block Models (SBM), whose general idea is that the nodes are partitioned into groups and the probability that two nodes are connected depends on the groups to which they belong [5]. The third class of methods is model-free and includes machine learning algorithms for graph embedding. Such methods convert the graph data into a low dimensional space in which certain graph structural information and properties are preserved. Different graph embedding variants have been developed aiming to preserve different information in the embedded space, some examples are: HOPE [6], node2vec [7], NetSMF [8], ProNE [9]. A fourth category comprises neural message-passing models tailored for link prediction. Neural Common Neighbor with Completion (NCNC) [10] integrates structural features with message passing under an MPNN-then-SF architecture and corrects for graph incompleteness by completing missing common-neighbor structures. Finally, the Message Passing Link Predictor (MPLP) [11] approximates structural heuristics such as Common Neighbor via quasi-orthogonal vector propagation within a pure message-passing framework.

While the aforementioned approaches have been shown to be competitive link predictors, they do not offer a clear interpretability of the mechanisms behind the network growth. This property, instead, can be fulfilled for example by mechanistic models based on the Cannistraci-Hebb theory [3, 12–16], since each of them is based on an explicit deterministic mathematical formulation. Each model in the Cannistraci-Hebb theory represents a specific rule of self-organization which is associated to explicit principles that drive the growth's dynamics of the underlying complex networked physical system.

The concept of network automata, like other forms of automata, is rooted in AI research [17–19], where they model adaptive, decentralized, and emergent intelligence mechanisms in complex networks. The Cannistraci-Hebb theory is a recent achievement in network science [16, 20] that includes a theoretical framework to understand local-based link prediction on paths of length n under the lens of predictive network automata theory. CH theory goes beyond any type of classical local link predictor heuristic on paths of length two such common neighbors (CN) [21], resource allocation (RA) [22], Jaccard [23] and preferential attachment (PA) [21]; and link predictor on paths of length three of Kovács et al. [24], which triggered a fundamental discovery on the organization of protein interaction networks (PPI). Following the discovery of a previous article of Daminelli et al. [12] that stressed the importance of paths of length three for link prediction in bipartite networks, on the same line Kovács et al. suggested that proteins interact according to an underlying bipartite scheme. Indeed, proteins interact not if they are similar to each other, but if one of them is similar to the other's partners. This principle, such as the one proposed by Daminelli et al. [12], mathematically relies on network paths of length three (L3) [24], whereas most of the deterministic local based models previously developed were based on paths of length two (L2) [3]. These findings lead to a change of perspective in the field, highlighting the existence of different classes of networks whose patterns of interactions are organized either as L2 or L3. However, a conceptual limitation of the studies of Daminelli et al. [12] and Kovács et al. is that the L3-based link predictors developed [24] were not properly connected to already known principles of modelling, which prompted us to formulate and introduce a generalized theory.

In this study, we introduce four key innovations in the field of topological link prediction:

- **Minimization of external connectivity (CH paradigm).** We formalize a new principle within the Cannistraci-Hebb (CH) framework that emphasizes minimizing external local-community links (eLCL), leading to the introduction of two new models, CH3 and CH3.1.

- **Engineering the adaptive network automata learning machine CHA.** We design an adaptive intelligent machine CHA, that automatically learns from the network topology the most suitable CH rule and path length to model each network, using internal validation to guide selection. This adaptive modeling is the central innovation of the study. Crucially, our framework infers the physical principle that governs link formation: L2-based rules reflect homophilic interactions (similarity-driven), while L3-based rules capture synergistic interactions (diversity-driven cooperation). Thus, CHA is not a black-box scorer but an interpretable, mechanistic machine that recovers the effective rule explaining the prediction and governing the topological evolution directly from data. This bridges AI and network science, enabling both predictive power and scientific insights across physics domains.

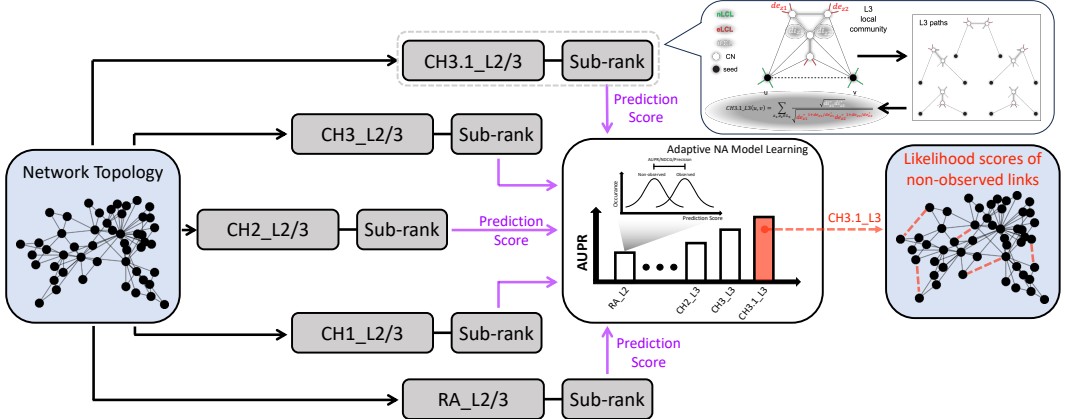

Figure 1: **Illustration of the Cannistraci-Hebb Adaptive (CHA) network automaton framework.** CHA is designed to automatically adapt to the structural characteristics of the input network by selecting the most suitable Cannistraci-Hebb (CH) model and path length for link prediction. For a given network, multiple CH models (e.g., CH2, CH3) and path lengths (e.g., $L2$, $L3$) are applied independently to assign likelihood scores to both observed and non-observed links. For each model-path combination, a performance metric, such as Area Under the Precision-Recall Curve (AUPR), Precision, or NDCG, is computed using observed links as positives. The combination achieving the highest value under the selected metric is chosen, and the corresponding scores for the non-observed links are used as the final prediction. To further refine the ranking among node pairs receiving identical CH scores, a sub-ranking strategy based on Spearman correlation of shortest-path profiles is applied. This adaptive procedure ensures that CHA selects the rule best aligned with the underlying connectivity pattern of the network under analysis.

Empirically, on a benchmark of over 1000 networks, CHA achieves more than **twice the win rate** of the best-performing baseline.

- **Comprehensive static and temporal benchmark.** We construct a large-scale benchmark ATLAS, consisting 1269 real-world networks (ATLAS-static) and 14 time-evolving networks (ATLAS-temporal).

- **Multi-metric evaluation.** We adopt three complementary evaluation metrics, Precision, NDCG, and AUPR, to capture diverse aspects of link prediction performance. Across all three metrics, our adaptive model **consistently outperforms** all baselines, demonstrating its robustness and general superiority under different evaluation criteria.

## 2 Preliminaries and Methods

### 2.1 Physical Modelling

#### 2.1.1 Network Automata

CH and CHA are network automata rules for approximating the likelihood of a non-observed link to appear in the network. These rules are categorized as network automata because they adopt only local information to infer the score of a link in the network without need of pre-training of the rule. Note that CH and CHA are predictive network automata that differ from generative network automata which are rules created to generate artificial networks [25–27]. Network automata as for any type of automata are part of AI research [17–19]. Network automata were originally introduced by Wolfram [28] and later formally defined by Smith et al. [29] as a general framework for modeling the evolution of network topology. Given an unweighted and undirected adjacency matrix $X(t)$ at time $t$, in a network automaton the states of links evolve over time according to a rule that depends only on local topological properties computable from a portion of the adjacency matrix $\tilde{X}(t) \subset X(t)$:

$$\tilde{X}(t+1) = F(\tilde{X}(t)) \tag{1}$$

The ruleset may depend on any property of the nodes or links and might be deterministic or stochastic. In contrast to cellular automata on a network [28, 30], in which the states of nodes evolve and

whose neighborhoods are defined by the network, in network automata the states of links evolve, and therefore the topology itself changes over time.

Smith et al. [29] provide an example in which the state update of a link $X_{u,v}(t+1)$ is determined by a simple topological property such as the sum of the node degrees:

$$f_{u,v}(t) = d_u(t) + d_v(t) \tag{2}$$

If the link exists, $X_{u,v}(t) = 1$, and the property exceeds a survival threshold $f_{u,v}(t) > x_S$, then the link survives; otherwise, it is removed. If the link does not exist, $X_{u,v}(t) = 0$, and the property exceeds a birth threshold $f_{u,v}(t) > x_B$, then the link is created; otherwise, it remains absent. In this example, the computation of the topological property and the link update based on the survival and birth thresholds constitute the $F$ operation. This basic ruleset fully describes the evolution of a network automaton.

We note that if we focus on the topological property $f_{u,v}(t)$, we could replace the sum of node degrees with any of several mathematical models developed for link prediction, such as common neighbors (CN) [21], resource allocation (RA) [22], Jaccard [23], and preferential attachment (PA) [21]. Although these models are often referred to as heuristics, the definition and example above clearly show that such local and deterministic models are in fact network automata.

As a final remark, in this link prediction study we specifically use these algorithms to predict the non-observed links that are more likely to be created at the next step of network evolution. Therefore, we do not consider the survival and birth thresholds in further detail, and focus solely on the topological property $f_{u,v}(t)$. For simplicity, we will also omit the time variable $t$ in the following discussion.

### 2.1.2 Network Automata on Paths of Length $n$

After having recalled the framework of network automata defined by Smith et al. [29], we now introduce a particular subclass named *network automata on paths of length* $n$. These automata evaluate the topological property between two nodes based on the topological information contained along the paths of length $n$ between them. In mathematical terms, we can express the topological property as follows:

$$f(u,v) = \sum_{z_1,\ldots,z_{n-1} \in L_n} f'(z_1,\ldots,z_{n-1}) \tag{3}$$

where $u$ and $v$ are the two seed nodes of the candidate interaction; the summation is executed over all paths of length $n$; $z_1,\ldots,z_{n-1}$ are the intermediate nodes on each path of length $n$; and $f'(z_1,\ldots,z_{n-1})$ is some function dependent on the intermediate nodes.

A simple example is represented by the *resource allocation* (RA) model developed by Zhou et al. [22], which is a network automaton on paths of length two ($L2$), using as function $f'(z)$ the inverse of the degree of the intermediate node (common neighbour in the $L2$ case). The mathematical formula is:

$$\text{RA-L2}(u,v) = \sum_{z \in L_2} \frac{1}{d_z} \tag{4}$$

where the summation is over all paths of length two; $z$ is the intermediate node on each path; and $d_z$ is the degree of node $z$.

To generalize this to paths of length $n > 2$, we need an operator that merges the individual topological contributions of the intermediate nodes on a path of length $n$. Without loss of generality, if we use the geometric mean as the merging operator, we derive the following generalized formula for RA on paths of length $n$:

$$\text{RA-Ln}(u,v) = \sum_{z_1,\ldots,z_{n-1} \in L_n} \frac{1}{\left(d_{z_1} d_{z_2} \cdots d_{z_{n-1}}\right)^{\frac{1}{n-1}}} \tag{5}$$

where the summation is executed over all paths of length $n$; $z_1,\ldots,z_{n-1}$ are the intermediate nodes; and $d_{z_1},\ldots,d_{z_{n-1}}$ are their respective degrees.

We note that for paths of length three ($L3$), the above formula becomes equivalent to the one proposed by Kovács et al. [24], which extends the resource allocation principle to paths of length three—although this connection was not properly clarified in their study, but was subsequently explained in [16] supporting the present one. From here onward, we will refer to this method as *RA-L3*.

### 2.1.3 Cannistraci-Hebb Network Automata on Paths of Length $n$

In this section, we introduce a new rule of self-organization that can be modeled using different network automata on paths of length $n$. Let us first recall some theoretical background.

In 1949, Donald Olding Hebb proposed a local learning rule in neuronal networks, often summarized as: *neurons that fire together wire together* [31]. However, the concept of "wiring together" was not fully specified and can be interpreted in two ways. The first interpretation is that existing connectivity between co-firing neurons is reinforced. The second is that new connections form between co-firing neurons not yet directly connected but already integrated within the same interacting cohort.

In 2013, Cannistraci et al. [3] termed the second interpretation *epitopological learning*, noting that it could be formalized as a topological link prediction problem in complex networks. The rationale is that, in networks with local-community organization, cohorts of neurons tend to be co-activated (fire together) and to learn by forming new connections (wire together), because they are topologically isolated within the same local community.

Cannistraci et al. [3] postulated that epitopological learning in neuronal networks is a special case of a more general rule of local learning, valid for topological link prediction in any complex network exhibiting a *local-community-paradigm* (LCP) architecture. Based on this idea, they introduced a new class of link predictors which outperformed state-of-the-art predictors in both monopartite [3, 32–38] and bipartite topologies [12, 13], across various domains such as brain networks, social networks, biological systems, and economic structures.

A study by Narula et al. [39] also highlighted that LCP and epitopological learning enhance the understanding of local brain connectivity in processing, learning, and memorizing chronic pain.

Previous formulations of the LCP theory emphasized the contribution of common neighbor nodes complemented by the interactions among them, termed internal local-community links (iLCL). This was a limitation, as shown by Cannistraci [14], who demonstrated that the local isolation of common neighbors, minimizing their interactions external to the local community (external local-community links, eLCL), is equally important to carve the LCP architecture. This minimization forms a topological energy barrier that confines information processing within the community.

Here, we introduce the **Cannistraci-Hebb (CH)** theory, a revised mathematical formalization of the LCP theory. The Cannistraci-Hebb (CH) theory provides a topological formulation of Hebbian learning, grounded on two pillars: (1) the minimization of external links (eLCL) within local communities, and (2) the path-based definition of local communities that capture homophilic (similarity-driven) interactions via paths of length 2 and synergetic (diversity-driven) interactions via paths of length 3. For any network automata rule, the necessary condition to be a CH rule is that explicitly incorporates the minimization of eLCL. We define *Cannistraci-Hebb network automata on paths of length $n$* as any network automaton in which the function $f'(z_1, \ldots, z_{n-1})$ follows the CH rule.

The first CH model, introduced by Cannistraci et al. [3] and originally named *Cannistraci-Resource-Allocation (CRA)*, is renamed here as CH1. Its formula for $L2$ is:

$$\text{CH1-L2}(u, v) = \sum_{z \in L_2} \frac{di_z}{d_z} \tag{6}$$

where $di_z$ is the internal degree (number of iLCL) of the intermediate node $z$, and $d_z = 2 + \text{iLCL} + \text{eLCL}$ is the total degree. This model encourages minimization of eLCL, but only when $di_z > 0$, otherwise the node does not contribute to the sum.

To address this, Cannistraci et al. [16] proposed the second CH model:

$$\text{CH2-L2}(u, v) = \sum_{z \in L_2} \frac{1 + di_z}{1 + de_z} \tag{7}$$

where $di_z$ and $de_z$ are the internal and external degrees (iLCL and eLCL) of node $z$, respectively. The unitary terms in the numerator and denominator prevent saturation when either value is zero.

Next, we introduce **CH3**, a novel model proposed for the first time in this study, which mathematically represents the basic principle of being a CH rule. Indeed, CH3 is based solely on eLCL minimization:

$$\text{CH3-L2}(u, v) = \sum_{z \in L_2} \frac{1}{1 + de_z} \tag{8}$$

CH3 departs from earlier formulations by fully discarding the internal degree component and focusing purely on penalizing external connectivity, providing a clean and principled expression of the CH paradigm.

Finally, we propose **CH3.1**, which embodies an adaptive mechanism: it accounts for the reward of internal links $di_z$ when the node $z$ follows the CH principle (because its number of external links $de_z$ is low), and progressively neglects the reward $di_z$ for larger values of external links $de_z$, which indicates a violation of the CH principle:

$$\text{CH3.1-L2}(u,v) = \sum_{z \in L_2} \frac{1 + di_z}{(1 + de_z)^{1 + \frac{de_z}{1 + de_z}}} \tag{9}$$

We note that the RA model is also a CH network automaton, while CN is not. In this study, we focus on five CH models: RA, CH1, CH2, CH3 and CH3.1. We do not consider non-CH models, including those studied by Kovács et al. [24], as they were shown to underperform.

Similar to RA, the CH models can be generalized to paths of length $n$ ($L_n$). Their formulas for $L2$, $L3$, and $Ln$ are summarized in Figure 4. In the generalized case, the local community is the set of all intermediate nodes in any path of length $n$ between the seed nodes. iLCL are the internal links among those nodes; eLCL are the links between any intermediate node and external nodes (excluding the seed nodes).

### 2.1.4  CH Model Sub-Ranking Strategy

Here we describe the sub-ranking strategy adopted by the CH model to internally sub-rank all node pairs that receive the same CH score. The goal is to refine link prediction by reducing the ranking uncertainty among node pairs that are tied-ranked.

Given a network and a set of CH scores $\text{CH}_{i,j}$ computed for all node pairs $(i,j)$ according to a given CH model, the sub-ranking procedure proceeds as follows:

1. Assign to each link $(i,j)$ in the network a weight $w_{i,j} = 1/\text{CH}_{i,j}$ to transform similarity into dissimilarity.

2. Compute the shortest paths (SP) between all node pairs in the resulting weighted network.

3. For each node pair $(i,j)$, compute the prediction score $\text{SPcorr}_{i,j}$ as the Spearman's rank correlation between the two vectors of all shortest paths from node $i$ and from node $j$ to every other node in the network.

4. Generate a final ranking of node pairs such that pairs are first ranked by $\text{CH}_{i,j}$, and any ties are sub-ranked using $\text{SPcorr}_{i,j}$. If both scores are tied, then the node pairs receive the same final rank.

5. (Optional) Map the final ranking back to a likelihood score if a numerical prediction score is required by downstream applications (see details in Appendix F).

Although the SPcorr score could be replaced by other link predictors, we chose this approach for its neurobiological grounding, which aligns with the CH model's conceptual framework. Specifically, based on one interpretation of *Peters' rule*, the probability of two neurons being connected is proportional to the spatial apposition of their respective axonal and dendritic arbors [40]. In other words, connectivity depends on the geometrical proximity of neurons.

This biological principle resonates with the SPcorr score within the CH modelling framework: a high correlation implies that two nodes share similar shortest-path distances to all other nodes, which suggests, within the network topology, that they are spatially proximate due to their network-geometric closeness.

### 2.2  Engineering the Adaptive Network Automata Machine

Different types of complex networks exhibit distinct structural patterns, some are better captured by $L2$ connectivity, others by $L3$, making it unlikely that a single network automaton rule can perform optimally across all domains (see Figure 3). For instance, $L2$-based rules may suit social networks, whereas protein–protein interaction (PPI) networks often favor $L3$-based approaches.

Here, we aim to go a step further from the engineering perspective and design a computational machine that is adaptive to the network under investigation and is capable of automatically selecting the model that is most likely to provide the best prediction.

To achieve this, we exploit a particular property of the network automata models discussed in Section 2.1. These deterministic rules for link prediction assign scores to both observed and non-observed links in a way that is directly comparable; that is, the scores of observed links are not inherently biased to be higher or lower than those of non-observed links. This is because the mathematical formulation used to compute the score between two nodes is independent of the existence of the link in the current topology.

Specifically, given a set of candidate models and a network, we compute a metric (e.g., AUPR, Precision, or NDCG) based on how well each model separates observed from non-observed links. The assumption is that, if the model tends to assign higher scores to observed links than to non-observed links, it is likely more effective at predicting missing or future links.

This mechanism enables CHA to self-adapt to a wide range of network topologies. The time complexity and runtime details are provided in Appendix E.

## 3 Experiments

### 3.1 Datasets and Baselines

**Datasets.** We evaluate our methods on a comprehensive collection of real-world networks drawn from two benchmark sets:

- **ATLAS-static** includes 1269 undirected static networks from 14 domains such as biological, social, and economic systems (see Appendix for full details).
- **ATLAS-temporal** consists of 14 real-world networks with temporal snapshots representing dynamic evolution across time (see Supplementary material for details).

Table 1: Number of real-world networks tested for the link prediction task in related literature.

| Algorithm | Field | Year | Networks | Ref. |
|---|---|---|---|---|
| SBM | Statistical Physics | 2014 | 8 | [41] |
| SBM-DC | Statistical Physics | 2014 | 5 | [42] |
| SBM-N, SBM-DC-N | Statistical Physics | 2014 | 33 | [5] |
| SPM | Quantum Physics | 2015 | 13 | [4] |
| HOPE | Computer Science | 2016 | 4 | [6] |
| node2vec | Computer Science | 2016 | 3 | [7] |
| ProNE, ProNE-SMF | Computer Science | 2019 | 5 | [9] |
| NetSMF | Computer Science | 2019 | 5 | [8] |
| MPLP, MPLP+ | Computer Science | 2024 | 15 | [11] |
| **CHA** | **Physics & CS** | **2025** | **1283** | **Ours** |

For static networks, we adopt a 10% link removal evaluation protocol: 10% of existing edges are randomly removed from the original network and used as positives in a held-out test set, while the remaining links form the input to the prediction algorithm. The evaluation is repeated 10 times with different random splits, and the average metric is reported. For temporal networks, we evaluate predictions across successive snapshots: links that appear at future time steps are used as positives, while links absent at prediction time are ranked and scored. The average performance across all time pairs is reported. Full evaluation protocols are described in Appendix B.

**Baselines.** We compare the proposed CHA framework against four categories of state-of-the-art link prediction methods:

- **SPM** (Structural Perturbation Method) [4]: a model-free global approach based on spectral perturbation.
- **SBM variants** [5, 41, 42]: including SBM, degree-corrected SBM, and nested extensions.
- **Graph Embedding Methods**: including HOPE [6], node2vec [7], NetSMF [8], and ProNE [9].

- **Message-Passing Graph Neural Networks**: including NCNC [10], MPLP [11], MPLP+, and MPLP+A. NCNC combines message passing with structural features under the MPNN-then-SF architecture and performs graph completion to mitigate incompleteness. MPLP and MPLP+ approximate classical heuristics such as Common Neighbor through quasi-orthogonal message propagation. The new variant **MPLP+A** adaptively selects between the L2-based (homophilic) and L3-based (synergetic) versions of MPLP+ according to validation performance.

Detailed descriptions of all baseline methods, including their underlying principles, implementation details, and hyperparameter settings, as well as complete evaluation procedures, are provided in the Appendix A.

**Scale of Evaluation.** Unlike prior work that typically evaluates on a limited number of networks (often fewer than 20), our study conducts large-scale benchmarking on a total of 1283 real-world networks (1269 static and 14 temporal). This represents the most extensive evaluation to date for the link prediction task. Table 1 summarizes the number of networks used in related literature, highlighting the comprehensiveness of our experimental setup.

## 3.2 Link Prediction on ATLAS-static

To evaluate robustness across network scales, we define three nested subsets of the ATLAS-static dataset: **ATLAS-small** ($N \leq 100$, 900 networks), **ATLAS-medium** ($N \leq 2500$, 1126 networks), and **ATLAS-large** ($N \leq 10000$, 1269 networks). As shown in Figures 2, CHA consistently achieves both the highest win rate and the best average rank across all three settings, outperforming SPM, all SBM variants, graph embedding methods (HOPE, node2vec, NetSMF, ProNE), and message-passing models (NCNC, MPLP and MPLP+). Notably, CHA achieves more than **twice the win rate** of other baselines under AUPR, underscoring the strength of its adaptive mechanism. These results highlight CHA's superiority not only in winning the top position more frequently but also in maintaining consistently strong performance across networks of varying size and structure.

Additional mean-rank and win-rate comparisons based on NDCG and Precision are also reported in Figures 6 and 7, confirming the consistency of CHA's superiority under multiple evaluation metrics.

Notably, the adaptive variant MPLP+A achieved **better** average rank and higher win rate than MPLP+, confirming that adaptivity between L2 (homophilic) and L3 (synergetic) path rules can also benefit message-passing models. This observation supports the generality of the Cannistraci-Hebb theory beyond the CHA framework.

## 3.3 Temporal Link Prediction

We further evaluate the generalization of CHA in time-evolving settings using 14 real-world temporal networks from the ATLAS-temporal collection. Figures 2g and 2h report the performance of CHA compared to baseline methods in terms of mean rank and win rate based on AUPR, respectively. CHA ranks among the top-performing methods and achieves the best mean rank and highest win rate. These results indicate that CHA is well-suited for modeling dynamic connectivity in temporal networks, offering stable and competitive predictions across diverse time-evolving settings.

## 3.4 Path Length Preference Across Network Classes

To better understand the need for adaptivity in CHA, we analyze how different network classes prefer different path lengths. For each network in the ATLAS-static dataset ($N \leq 10000$), and for each CH model (RA, CH1, CH2, CH3, CH3.1), we apply the 10% link removal protocol and compute AUPR on both $L2$ and $L3$ paths. For each network and path length, we record the maximum AUPR across all CH models.

Figure 3 reports, for each network class, the win rate of $L2$ versus $L3$, i.e. how often one path length outperforms the other within the class. The results clearly show that no single path length dominates across all classes: some network types (e.g., coauthorship or connectome) are better captured by $L2$ structures, while others (e.g., PPI or transcription) favor $L3$.

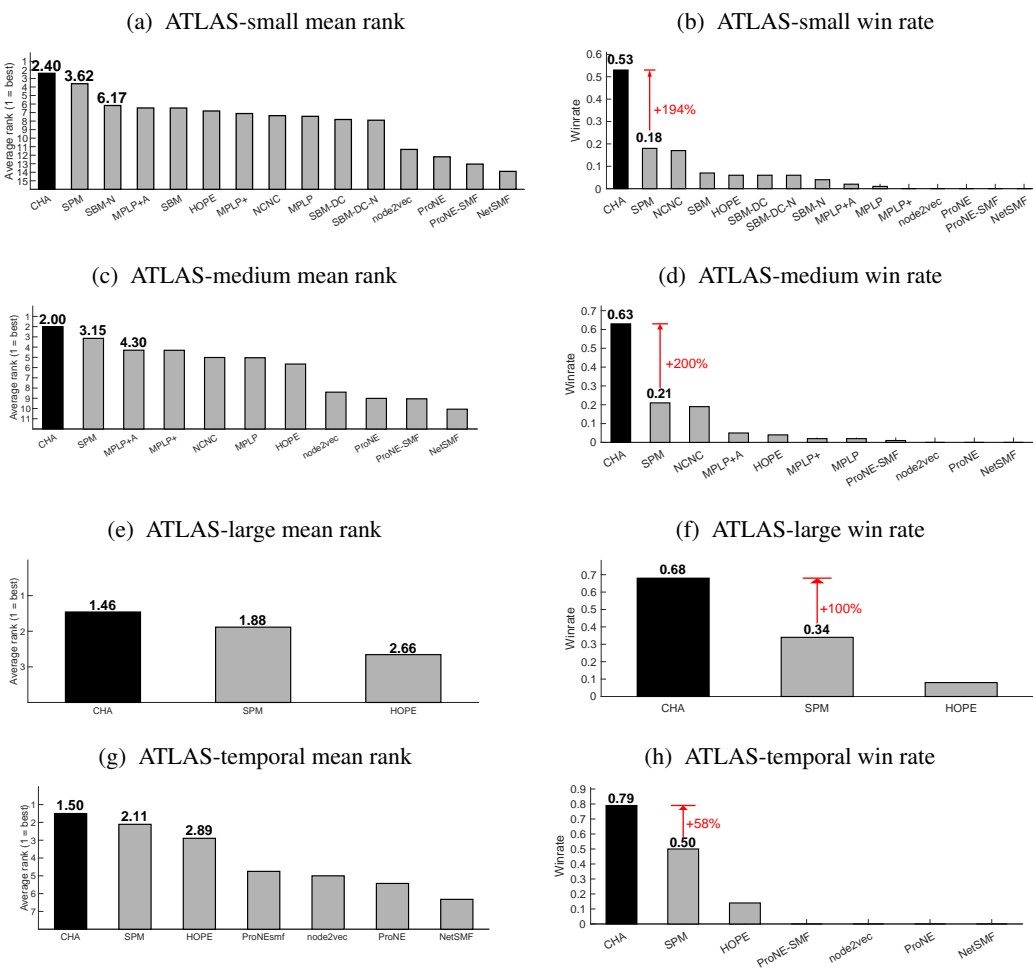

Figure 2: **Comparison of CHA with baseline methods across network categories using AUPR.** Each pair of subplots compares the performance of CHA and baselines in terms of **(left)** mean rank and **(right)** win rate across: **(a, b)** ATLAS-small ($N \leq 100$, 900 networks), **(c, d)** ATLAS-medium ($N \leq 2500$, 1126 networks), **(e, f)** ATLAS-large ($N \leq 10000$, 1269 networks), and **(g, h)** ATLAS-temporal (14 time-evolving networks). CHA consistently achieves the best mean rank and highest win rate across all categories.

This observation provides empirical motivation for using an adaptive mechanism that automatically selects the most suitable path length and CH model for each network, rather than relying on a fixed configuration. Similar trends are observed when using Precision and NDCG as evaluation metrics (see Appendix Figures 8 and 9). A related experiment on *ogbl-collab* (Appendix G) further confirms that other message-passing models such as MPLP+ exhibit a similar L2–L3 preference pattern, consistent with the adaptive principles of CH theory.

### 3.5 Validation of the CH Adaptive Strategy

The CHA framework operates by adaptively selecting, for each network, the CH model and path length combination that achieves the best predictive performance on observed links. In this study, we instantiate CHA using the CH3 and CH3.1 models, as they best capture the core CH principle of minimizing external links (eLCL).

To directly assess the value of the adaptive mechanism in CHA, we compare its performance with all individual CH variants (RA, CH1, CH2, CH3, CH3.1) under the same 10% link removal evaluation

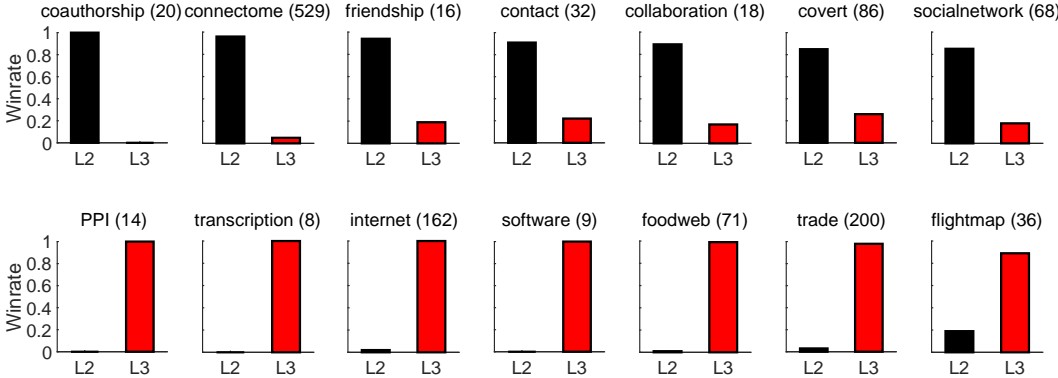

Figure 3: For each network of the ATLAS (considering $N \leq 10000$, 1269 networks) and for each CH model (RA, CH1, CH2, CH3, CH3.1) of path lengths L2-L3, we applied the 10% link removal evaluation, obtaining the AUPR as measure of performance . For each path length L2-L3, we assigned as performance on a network the maximum AUPR over the CH models. The barplots report for each network class and for each path length the win rate over the networks of that class. For each class, the number of networks is shown in brackets.

on the full ATLAS-static dataset ($N \leq 10000$, 1269 networks). For each network, we identify the winning method per metric, and report the average win rate across the 14 network classes.

Figure 5 shows results under three metrics: AUPR, Precision, and NDCG. In all cases, CHA outperforms every individual CH variant, consistently achieving the highest win rate. This confirms the effectiveness of the adaptive design in CHA, which dynamically selects the best CH model and path length combination for each network rather than relying on a fixed configuration.

We also compare this choice against other combinations of CH models and report the win rate of each configuration on the ATLAS-static dataset. As shown in Table 2, the {CH3, CH3.1} setting achieves the highest win rate, confirming its effectiveness as the optimal rule set for CHA.

## 4    Conclusion and Discussion

We proposed **Cannistraci-Hebb Adaptive (CHA)**, an adaptive network automata machine that exploits principles of network topological self-organization for link prediction. CHA is based on the CH theory, a topological formulation of Hebbian learning, grounded on two pillars: (1) the minimization of external links within local communities, and (2) the path-based definition of local communities that capture homophilic (similarity-driven) interactions via paths of length 2 and synergetic (diversity-driven) interactions via paths of length 3. CHA leverages two models, CH3 and CH3.1, and adaptively selects the optimal path length per network to capture local community Cannistraci-Hebbian driven topological dynamics and organization. Experiments on more than 1000 real networks, static and temporal, show that CHA consistently outperforms state-of-the-art baselines across multiple metrics. Bridging the physics of complex networks and artificial intelligence via adaptive network automata, this study confirms the effectiveness of combining theoretical grounding with adaptive engineering design.

While CHA is deterministic and interpretable, it does not leverage node attributes, which may be crucial in some domains. Future extensions could integrate topological and feature-based signals. CHA can support applications such as recommender systems or biological discovery, but its use on sensitive data should be carefully monitored to avoid unintended inferences.

## Acknowledgements

This work was supported by: The Zhou Yahui Chair Professorship award of Tsinghua University (to CVC); The National High-Level Talent Program of the Ministry of Science and Technology of China (grant number 20241710001, to CVC). The Center for Information Services and High Performance Computing (ZIH) of the TU Dresden and ScaDS.AI for providing HPC resources.

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

# Appendix

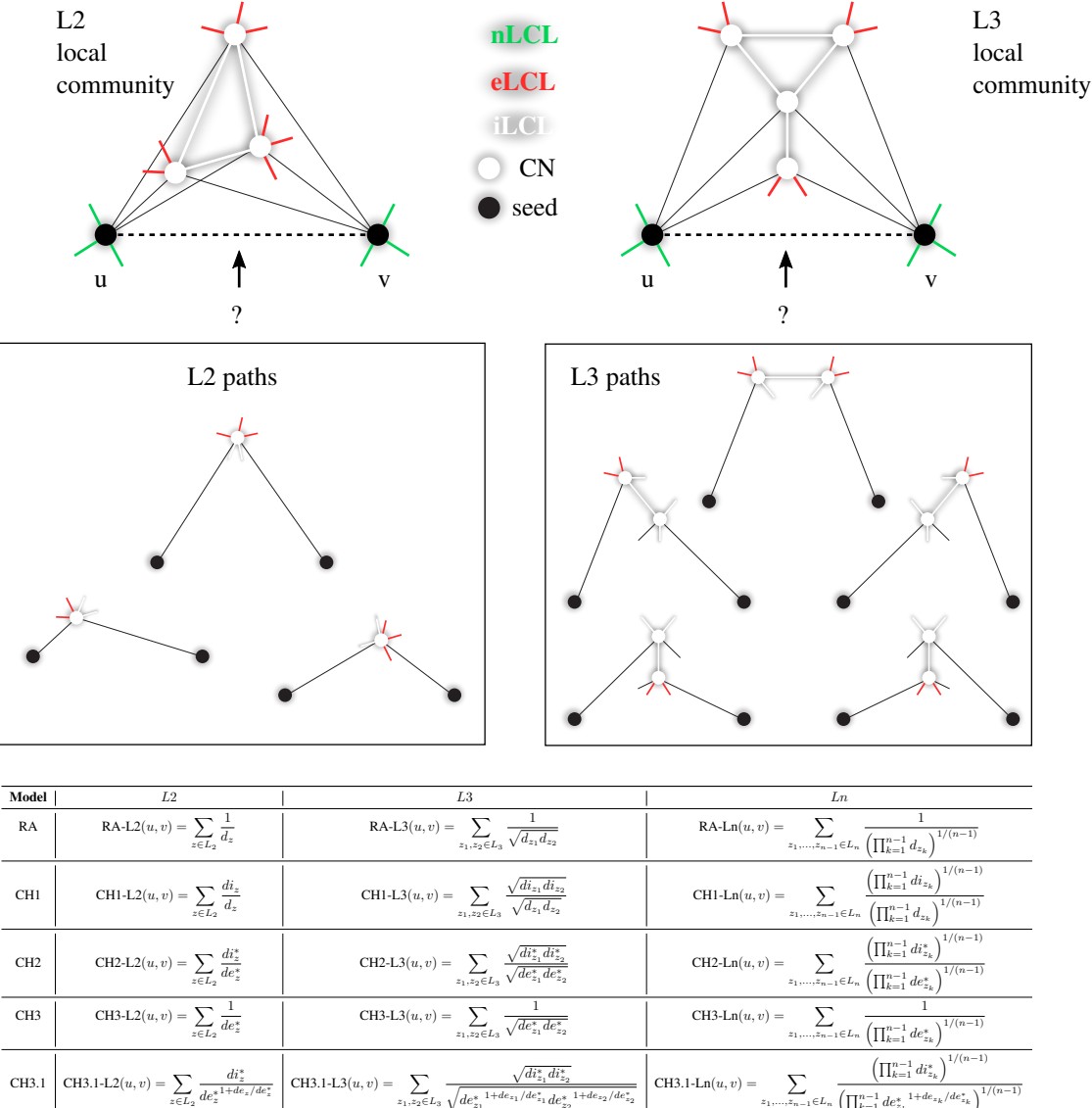

Figure 4: **Cannistraci-Hebb epitopological rationale.** The figure shows an explanatory example for the topological link prediction performed using the L2 or L3 Cannistraci-Hebb epitopological rationale. The two black nodes represent the seed nodes whose non-observed interaction should be scored with a likelihood. The white nodes are the L2 or L3 common-neighbours (CNs) of the seed nodes, further neighbours are not shown for simplicity. The cohort of common-neighbours and the iLCL form the local community. The different types of links are reported with different colours: non-LCL (green), external-LCL (red), internal-LCL (white). The set of L2 and L3 paths related to the given examples of local communities are shown. At the bottom, the mathematical description of the L2, L3 and Ln methods considered in this study are reported. Notation: u,v are the seed nodes; z is the intermediate node (CN) in the L2 path; $d_z$ is the degree of z; $di_z$ is the internal degree (number of iLCL) of z; $de_z$ is the external degree (number of eLCL) of z. For any degree it is valid the following: $d^* = 1 + d$. For L3 and Ln paths the definitions are analogous.

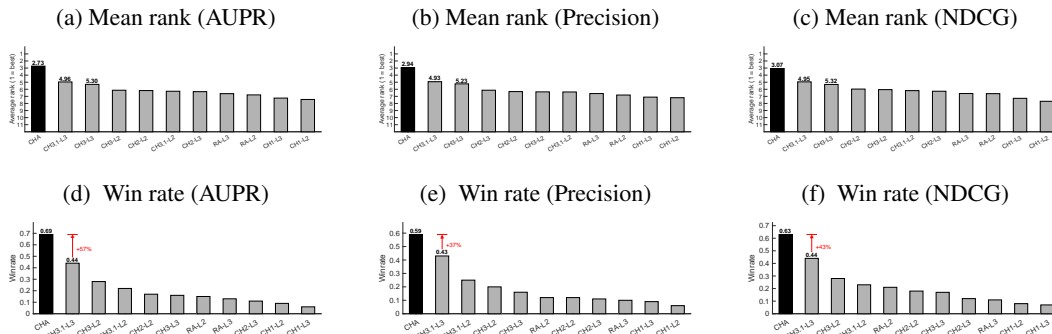

Figure 5: **Comparison of CHA and CH variants across evaluation metrics.** For each network in ATLAS-static ($N \leq 10000$, 1269 networks), we apply the 10% link removal evaluation and compute both **mean rank** (top row) and **win rate** (bottom row) across three metrics: (a, d) AUPR, (b, e) Precision, and (c, f) NDCG. Bars report average values over the 14 network classes. In all cases, CHA achieves the best mean rank and highest win rate, consistently outperforming all individual CH variants.

# A    Link Prediction Methods

## A.1    Structural Perturbation Method (SPM)

The Structural Perturbation Method (SPM) is based on a theory analogous to the first-order perturbation technique in quantum mechanics [4]. A high-level description of the procedure is as follows:

1. Randomly remove 10% of the links from the network adjacency matrix $X$, obtaining a reduced network $X' = X - R$, where $R$ is the set of removed links.

2. Compute the eigenvalues and eigenvectors of $X'$.

3. Considering the set of links $R$ as a perturbation to $X'$, construct the perturbed matrix $X_P$ via a first-order approximation that allows the eigenvalues to change while keeping the eigenvectors fixed.

4. Repeat steps 1–3 for 10 independent iterations and take the average of the resulting perturbed matrices $X_P$.

The link prediction result is given by the values in the averaged perturbed matrix, which represent the scores assigned to the non-observed links. A higher score indicates a higher likelihood that the corresponding interaction exists.

The rationale behind the method is that a missing portion of the network is predictable if its absence does not significantly alter the structural characteristics of the observable part, as captured by the eigenvectors of the adjacency matrix. If this condition holds, the perturbed matrices should serve as good approximations of the original network [4].

## A.2    Stochastic Block Model (SBM)

The general idea behind the *Stochastic Block Model* (SBM) is that the nodes of a network are partitioned into $B$ blocks, and a $B \times B$ matrix specifies the probabilities of links existing between nodes belonging to each pair of blocks. SBM provides a general framework for statistical analysis and inference in networks, particularly for tasks such as community detection and link prediction [41].

The concept of *degree-corrected* (DC) SBM was introduced for community detection [43] and for predicting spurious and missing links [42], in order to keep into account the variations in node degree typically observed in real networks. A *nested* (N) version of SBM has been introduced [5] to overcome two major limitations: the inability to separate true structures from noise, and the difficulty in detecting smaller yet well-defined clusters as network size increases.

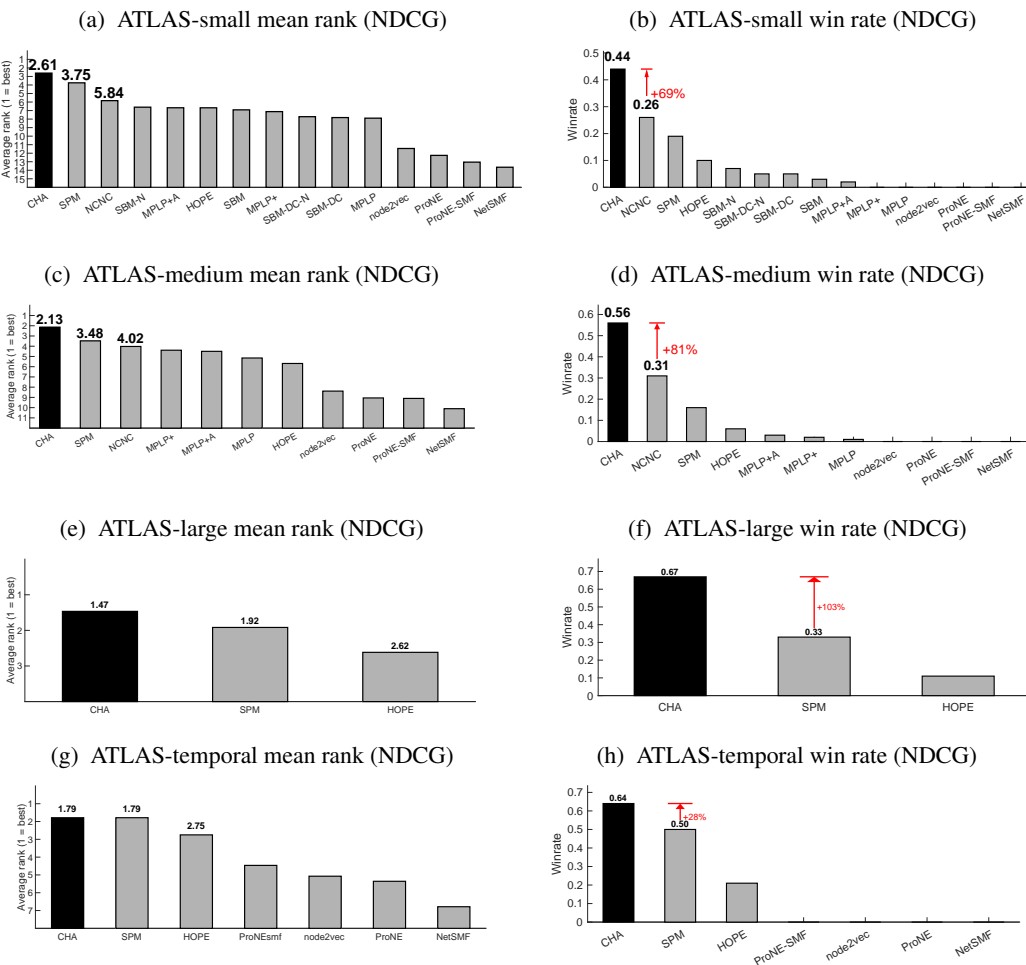

Figure 6: **Comparison of CHA with baseline methods across network categories using NDCG.**

All four tested variants (SBM, SBM-DC, SBM-N, SBM-DC-N) require finding an appropriate partitioning of the network to perform inference. We used the implementation provided in `Graph-tool` [44], which uses an optimized Markov Chain Monte Carlo (MCMC) algorithm to sample the space of possible partitions [41]. `Graph-tool` is a Python module available at: `http://graph-tool.skewed.de/`.

As suggested in [45], predictive performance is generally higher when averaging over multiple partitions rather than relying on a single most plausible partition, since the latter approach can lead to overfitting. Therefore, for each network, we sampled $P$ partitions, computed the likelihood scores for the non-observed links in each partition, and averaged the scores across all partitions to obtain the final link prediction result. We set $P = 100$ for ATLAS networks with $N \leq 100$, and $P = 50$ for connectomes with $N > 100$.

### A.3 HOPE

*High-Order Proximity preserved Embedding* (HOPE) is a graph embedding algorithm designed to preserve high-order proximities in graphs and to capture asymmetric transitivity [6]. Asymmetric transitivity depicts the correlation among directed edges, making HOPE particularly suitable for embedding directed networks, although it can also be used for undirected networks.

Many high-order proximity measures can reflect asymmetric transitivity in graphs, such as the *Katz index* [46]. Many of these measures share a common algebraic structure. Instead of computing the proximity matrix and then applying singular value decomposition (SVD), HOPE leverages this shared

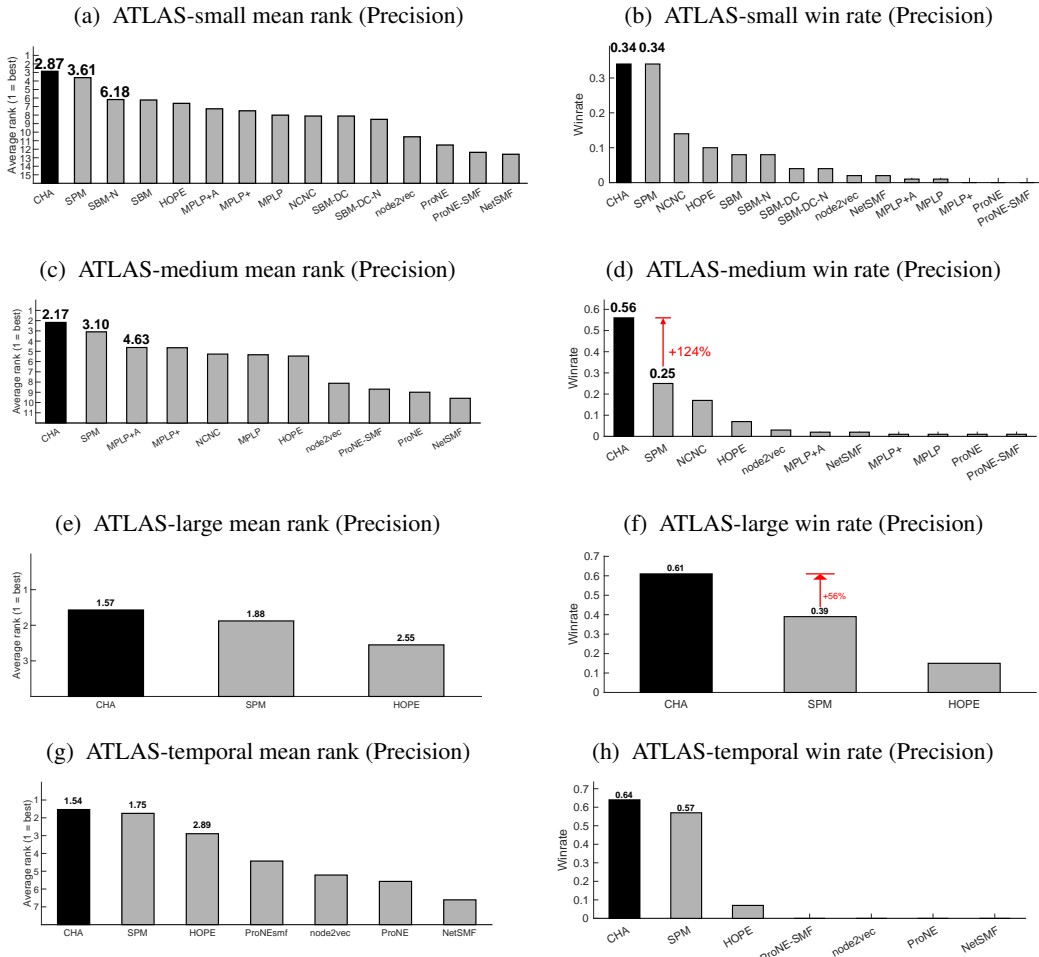

Figure 7: **Comparison of CHA with baseline methods across network categories using Precision.**

structure to transform the standard SVD problem into a *generalized SVD* problem. This formulation allows for the direct computation of embedding vectors, thereby avoiding explicit construction of the proximity matrix [6].

However, for the purpose of link prediction, an approximation of the proximity matrix can be reconstructed from the learned embedding. In this context, HOPE provides a scalable solution for approximating the Katz index through graph embedding. The entries of the approximated Katz proximity matrix represent the link prediction scores: the higher the proximity, the more likely the existence of the interaction.

The implementation of HOPE is available at: `https://github.com/ZW-ZHANG/HOPE`. For our experiments, we set the embedding dimension to $\min(128, N)$, where $N$ is the number of nodes, and used the default values for all other parameters.

### A.4 node2vec

*node2vec* is a graph embedding algorithm that maps nodes to a low-dimensional feature space by maximizing the likelihood of preserving the network neighborhoods of nodes [7]. The maximization is performed on a custom graph-based objective function using stochastic gradient descent, inspired by prior work in natural language processing and related to the Skip-gram model [7].

To define node neighborhoods flexibly, node2vec employs a second-order random walk strategy to sample node neighborhoods. The behavior of the random walk is governed by two parameters: $p$

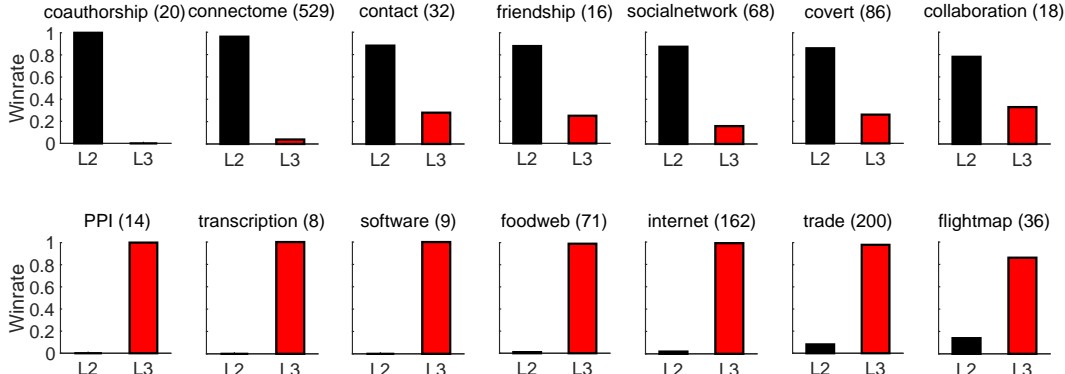

Figure 8: **Path length preference across network classes based on Precision.** For each network in ATLAS-static ($N \leq 10000$) and each CH model (RA, CH1, CH2, CH3, CH3.1), we compute Precision under 10% link removal. For each path length ($L2$, $L3$), we retain the best-performing CH model per network. Bar plots report the win rate of $L2$ versus $L3$ across networks in each class (class size in brackets).

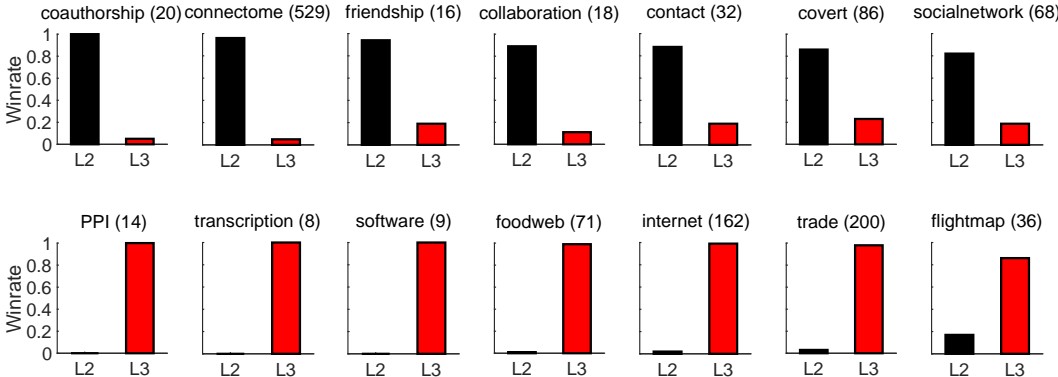

Figure 9: **Path length preference across network classes based on NDCG.** For each network in ATLAS-static ($N \leq 10000$) and each CH model (RA, CH1, CH2, CH3, CH3.1), we compute NDCG under 10% link removal. For each path length ($L2$, $L3$), we retain the best-performing CH model per network. Bar plots report the win rate of $L2$ versus $L3$ across networks in each class (class size in brackets).

(return parameter) and $q$ (in-out parameter), which bias the walk towards different network exploration strategies (e.g., breadth-first vs. depth-first search).

After computing the node embeddings, we generate feature vectors for node pairs by applying the Hadamard (element-wise) product to the embedding vectors of each node in the pair, as suggested in the original node2vec study [7]. These node-pair feature vectors are then used to train a logistic regression classifier, which outputs likelihood scores for the non-observed links in the network.

The implementation of the node2vec embedding method is available at: `https://github.com/snap-stanford/snap/`. We set the embedding dimension to $\min(128, N - 1)$, where $N$ is the number of nodes, and discarded node features that were constant across all nodes.

We tested the parameters $p$ and $q$ using three configurations: $(p = 0.5, q = 2)$; $(p = 1, q = 1)$; and $(p = 2, q = 0.5)$. The best configuration was selected via cross-validation. All other parameters were kept at their default values.

### A.5 ProNE and ProNE-SMF

*ProNE* has been proposed as a fast and scalable graph embedding algorithm that maps nodes to a low-dimensional feature space using a two-step procedure [9].

Table 2: **Results of CHA variants on ATLAS-static.** We evaluate several CHA variants on path lengths $L2$–$L3$, each using a different combination of CH models. For every network in ATLAS-static, we apply the 10% link removal evaluation and compute AUPR as the performance metric. For each network class, we report the **win rate** and **mean AUPR** of each algorithm across the networks in that class. The table shows the average of these values over all 14 classes. Algorithms are sorted by decreasing win rate. The upper bound indicates the performance that would be achieved by selecting the best CH model and path length per network.

| Method | Win rate | AUPR |
|---|---|---|
| upper bound | 1.00 | 0.30 |
| CH3-CH3.1 | 0.72 | 0.29 |
| CH2-CH3-CH3.1 | 0.72 | 0.29 |
| RA-CH2-CH3-CH3.1 | 0.70 | 0.29 |
| RA-CH3-CH3.1 | 0.69 | 0.29 |
| CH2-CH3.1 | 0.67 | 0.28 |
| CH3.1 | 0.65 | 0.28 |
| RA-CH2-CH3.1 | 0.62 | 0.28 |
| RA-CH3.1 | 0.60 | 0.28 |
| CH1-CH3-CH3.1 | 0.56 | 0.27 |
| CH1-CH2-CH3-CH3.1 | 0.56 | 0.27 |
| CH1-CH2-CH3.1 | 0.54 | 0.26 |
| RA-CH1-CH3-CH3.1 | 0.53 | 0.27 |
| RA-CH1-CH2-CH3-CH3.1 | 0.53 | 0.27 |
| CH1-CH3.1 | 0.52 | 0.26 |
| RA-CH1-CH2-CH3.1 | 0.48 | 0.26 |
| RA-CH1-CH3.1 | 0.47 | 0.26 |
| CH3 | 0.41 | 0.28 |
| CH2-CH3 | 0.41 | 0.28 |
| RA-CH3 | 0.38 | 0.28 |
| RA-CH2-CH3 | 0.38 | 0.28 |
| CH2 | 0.26 | 0.27 |
| RA-CH2 | 0.26 | 0.27 |
| CH1-CH2-CH3 | 0.26 | 0.26 |
| RA | 0.25 | 0.26 |
| CH1-CH3 | 0.25 | 0.26 |
| RA-CH1-CH2-CH3 | 0.23 | 0.25 |
| RA-CH1-CH3 | 0.22 | 0.25 |
| CH1-CH2 | 0.16 | 0.25 |
| RA-CH1-CH2 | 0.12 | 0.25 |
| RA-CH1 | 0.12 | 0.24 |
| CH1 | 0.10 | 0.23 |

The first step initializes the network embedding using sparse matrix factorization (SMF), which efficiently provides an initial node representation via randomized truncated singular value decomposition. The second step, inspired by the higher-order Cheeger's inequality, performs *spectral propagation* to enhance the initial embedding [9].

In our analysis, we considered both the embeddings obtained after the first step (*ProNE-SMF*) and those obtained after the second step (*ProNE*).

After computing the embeddings, we generated feature vectors for node pairs by applying the Hadamard product to the corresponding node embeddings. These node-pair features were then used to train a logistic regression classifier to produce likelihood scores for the non-observed links in the network.

The implementation of the ProNE and ProNE-SMF embedding methods is available at: `https://github.com/THUDM/ProNE/`. We set the embedding dimension to $\min(128, N-1)$, where $N$ is the number of nodes, and discarded node features that were constant across all nodes. Default values were used for all other parameters.

## A.6 NetSMF

*NetSMF* is a graph embedding algorithm that maps nodes to a low-dimensional feature space [8]. It is based on the observation that several network embedding algorithms implicitly factorize a specific closed-form matrix, and that explicitly factorizing this matrix can lead to improved performance. However, the matrix in question is typically dense, making it computationally expensive to handle for large networks.

NetSMF addresses this limitation by proposing a scalable solution that first applies spectral graph sparsification techniques to construct a sparse matrix that is spectrally close to the original dense matrix. It then performs randomized singular value decomposition (SVD) on the sparse matrix to efficiently obtain the node embeddings [8].

After generating the embeddings, we compute feature vectors for node pairs using the Hadamard product of their corresponding node embeddings. These pairwise feature vectors are then used to train a logistic regression classifier that produces likelihood scores for the non-observed links in the network.

The implementation of the NetSMF method is available at: `https://github.com/xptree/NetSMF/`. We set the embedding dimension to $\min(128, N-1)$, where $N$ is the number of nodes, and discarded node features with constant values across all nodes. We set `rounds = 10000` and used default values for all other parameters.

### A.7   Logistic Regression Classifier

After obtaining feature vectors for each node pair from the network embeddings generated by *node2vec*, *ProNE*, *ProNE-SMF*, and *NetSMF*, we trained a logistic regression classifier to compute likelihood scores for the non-observed links in the network.

In particular, we performed a repeated 5-fold cross-validation 10 times. For each repetition $i \in [1, 10]$, the following steps were executed:

1. Create a learning set consisting of all the observed links and an equal number of non-observed links (if available; otherwise, include all non-observed links).
2. Split the learning set into 5 folds for cross-validation.
3. For each cross-validation iteration $j \in [1, 5]$:
   (a) *Train:* Train a logistic regression classifier using 4 folds and obtain the coefficient estimates $B_{i,j}$.
   (b) *Validation:* Using the coefficients $B_{i,j}$, obtain the likelihood scores for the remaining fold and compute the prediction performance using $\text{AUPR}_{i,j}$.

After completing the 10 repetitions, we compute the mean coefficient estimates $\bar{B}$ across all $(i, j)$ pairs. These coefficients $\bar{B}$ are then used to compute the final likelihood scores for the non-observed links, which constitute the link prediction result.

In the case of *node2vec*, where multiple parameter configurations are tested, we also compute the mean validation performance $\overline{\text{AUPR}}$ over the 10 repetitions and 5 cross-validation iterations for each configuration. The final link prediction result corresponds to the configuration with the highest $\overline{\text{AUPR}}$.

In contrast, for *ProNE*, *ProNE-SMF*, and *NetSMF*, which use a single parameter configuration, step 3.(b) (validation) is not necessary.

We used the MATLAB implementation of the logistic regression classifier, specifically the `mnrfit` and `mnrval` functions, to perform model training and scoring.

### A.8   MPLP and MPLP+

*Message Passing Link Predictor (MPLP)* is a graph neural model specifically designed for the link prediction task [11]. Unlike general-purpose graph embedding methods that focus on node-level representations, MPLP explicitly estimates link-level structural features such as the Common Neighbor (CN) score. It achieves this by propagating quasi-orthogonal vectors through message-passing layers and leveraging their inner products to approximate structural similarities between node pairs.

To improve scalability, *MPLP+* introduces a more efficient variant that avoids expensive multi-hop preprocessing. Instead, it computes approximated structural signals using only one-hop neighborhoods, making it suitable for large-scale graphs with limited memory or time constraints.

The official implementation is available at: `https://github.com/Barcavin/efficient-node-labelling`. In this study, we followed the original MPLP experimental

settings, with the exception of the early stopping strategy. The original implementation used Hits@100 on a validation set with a 1:1 positive-to-negative ratio, which is not consistent with our evaluation metric, AUPR, where positives are ranked among all missing links. This misalignment could result in suboptimal model selection. To address this, we modified MPLP's early stopping criterion to monitor the AUPR on the validation set and stop training when it no longer improves, ensuring better alignment with our evaluation framework.

**Hyperparameter search for MPLP / MPLP+.** To ensure a fair comparison, we performed a targeted hyperparameter search on MPLP and MPLP+ varying the *batch size* $B \in \{128, 256, 512\}$ (other settings kept as in the official code). This follows the hyperparameter sensitivity emphasized in the original study.

**MPLP+A.** To further align with CHA's adaptive design, we introduce an additional baseline, *MPLP+A*, which automatically selects between the L2-only and L3-only versions of MPLP+ based on validation performance.

### A.9 NCNC

*Neural Common Neighbor with Completion (NCNC)* [10] is a recent neural link predictor that combines message passing with structural features under the MPNN-then-SF architecture. It enhances the original Neural Common Neighbor (NCN) model by completing missing common-neighbor structures to mitigate graph incompleteness before applying NCN on the completed graph. In our experiments NCNC failed to run on a small subset of networks (about 10%), and repeated runs consistently crashed; these failed cases were assigned the lowest rank in evaluation. For fairness, we also conducted a hyperparameter search over the batch size $B \in \{128, 256, 512\}$, selecting the best configuration for each dataset.

## B Link Prediction Evaluation

### B.1 10% Link Removal Evaluation

The 10% link removal evaluation framework is employed when there is no information available about missing links or links that may appear in the future relative to the current state of the network.

Given a network $X$, 10% of its links are randomly removed, resulting in a reduced network $X' = X - R$, where $R$ is the set of removed links. To evaluate a given algorithm, the reduced network $X'$ is provided as input, and the algorithm outputs likelihood scores for the non-observed links in $X'$.

These non-observed links are ranked in descending order of their predicted scores. The removed links $R$ are treated as positives, and the remaining non-links as negatives. Evaluation metrics such as area under the precision-recall curve (AUPR), Precision, and normalized discounted cumulative gain (NDCG) are computed to assess the ranking quality.

Because the link removal is random, the procedure is repeated 10 times with different train/test splits. The final performance on network $X$ is reported as the average metric over these repetitions.

### B.2 Temporal Evaluation

The temporal evaluation framework is employed when information is available regarding links that will appear in the future relative to the current time point of the network under consideration.

For a given network, a sequence of $T$ snapshots is available, each corresponding to a different time point. For each snapshot at time $i \in [1, T-1]$, the snapshot is provided as input to the algorithm being evaluated, which outputs likelihood scores for the non-observed links at time $i$.

For each pair of time points $(i, j)$ with $i \in [1, T-1]$ and $j \in [i+1, T]$, the non-observed links at time $i$ are ranked by their predicted scores, and the links that actually appear at time $j$ are treated as positives. Non-observed links at time $i$ involving nodes that no longer exist at time $j$ are excluded from the evaluation.

Multiple ranking-based metrics are computed for each $(i, j)$ pair, including area under the precision-recall curve (AUPR), Precision, and normalized discounted cumulative gain (NDCG). The final performance for the network is reported as the average of each metric over all valid time pairs.

## C  Datasets

### C.1  ATLAS

We have collected a dataset of 1269 real-world networks, either downloaded from publicly available online sources or provided directly by the authors of prior scientific studies. The networks have been categorized into 14 distinct classes, with the number of networks in each class

Table 3: Number of networks per class.

| Class | Count |
|---|---|
| Collaboration | 18 |
| Contact | 32 |
| Covert | 86 |
| Friendship | 16 |
| PPI | 14 |
| Connectome | 529 |
| Foodweb | 71 |
| Trade | 200 |
| Transcription | 8 |
| Coauthorship | 20 |
| Flightmap | 36 |
| Internet | 162 |
| Socialnetwork | 68 |
| Software | 9 |
| **Total** | **1269** |

For a complete list of the networks along with their basic properties (such as number of nodes and edges), references, data sources, and descriptions, please refer to Supplementary Material.

### C.2  Temporal Networks

We have collected a dataset of 14 real networks with temporal information, downloaded from publicly available online sources. For each network, a certain number of snapshots are available, corresponding to different time points.

For a complete list of the networks along with their basic properties (such as number of nodes and edges), references, data sources, and descriptions, please refer to Supplementary Material.

## D  Compute Resources

All experiments were conducted on a high-performance computing server equipped with 256 logical CPUs and 2 TB of RAM. The machine supports 64-bit architecture with 256 MiB of L3 cache and a base frequency of 1.5 GHz (boost up to 2.6 GHz). No GPUs were used, as CHA is CPU-based.

We estimate that running all 1269 networks required approximately 72 CPU hours.

## E  Time Complexity and Runtime Analysis

### E.1  Time Complexity of CHA

The time complexity of CHA is determined by the number of length-$\ell$ paths in the network and the cost of computing iLCL and eLCL statistics for the intermediate nodes along those paths.

Let $n$ and $m$ denote the number of nodes and edges, respectively. $\bar{d} = 2m/n$ is the average degree.

**For $\ell = 2$.**

- **Path count.** Each length-2 path $u \to z \to v$ is defined by an intermediate node $z$ connected to both $u$ and $v$. The total number of such paths is given by:

$$\text{\#L2\_path} = \sum_{z=1}^{n} \binom{d_z}{2} = \sum_{z=1}^{n} \frac{d_z(d_z - 1)}{2} = \mathcal{O}\left(\sum_{z=1}^{n} d_z^2\right)$$

where $d_z$ is the degree of node $z$. This represents the number of unique unordered two-hop paths in the network.

- **Computation per path.** For each length-2 path, CHA computes a score based on the iLCL and eLCL of the intermediate node $z$. This requires checking the neighbors of $z$ against the local community associated with the pair $(u, v)$, which takes $\mathcal{O}(d_z)$ time per path.

- **Overall time complexity.** Multiplying the path count and per-path cost gives the total time complexity:

$$\mathcal{O}\left(\sum_{z=1}^{n} d_z^2 \cdot d_z\right) = \mathcal{O}\left(\sum_{z=1}^{n} d_z^3\right)$$

We now analyze this quantity under three typical network regimes:

- *Sparse, degree-homogeneous:* If the graph is Sparse (i.e. $\bar{d} = 2m/n = \mathcal{O}(1)$) with relatively uniform degrees (i.e., $d_z = \mathcal{O}(1)$ for all $z$), then:

$$\mathcal{O}\left(\sum_{z=1}^{n} d_z^3\right) = \mathcal{O}(n)$$

So the overall time complexity of $\mathcal{O}(n)$.

- *Sparse, degree-heterogeneous:* If the graph is sparse (i.e., $\bar{d} = \mathcal{O}(1)$), but has a skewed degree distribution (e.g., power law), we can no longer assume $d_z = \mathcal{O}(1)$ for all nodes. To handle this case, we apply a relaxation via Hölder's inequality [47] to upper-bound the root-mean-cube degree $\left(\frac{1}{n} \sum_z d_z^3\right)^{1/3}$ in terms of the average degree:

$$\left(\frac{1}{n} \sum_{z=1}^{n} d_z^3\right)^{1/3} \leq n^{2/3} \cdot \left(\frac{1}{n} \sum_{z=1}^{n} d_z\right) = n^{2/3} \cdot \bar{d} = \mathcal{O}(n^{2/3})$$

This relaxation allows us to express the cubic-degree term in the overall complexity as:

$$\mathcal{O}\left(\sum_{z=1}^{n} d_z^3\right) = \mathcal{O}\left(n \cdot \left(\frac{1}{n} \sum_{z=1}^{n} d_z^3\right)\right) = \mathcal{O}\left(n \cdot \left(n^{2/3}\right)^3\right) = \mathcal{O}(n^3)$$

Thus, the overall time complexity in this case is $\mathcal{O}(n^3)$.

- *Dense graphs:* In the worst-case scenario of dense graphs, where $d_z = \mathcal{O}(n)$ for all nodes, we obtain:

$$\sum_{z=1}^{n} d_z^3 = \mathcal{O}(n^4)$$

leading to an overall time complexity of $\mathcal{O}(n^4)$.

**For $\ell = 3$.**

- **Path count.** Each length-3 path $u \to i \to j \to v$ passes through a central edge $(i, j) \in E$. The number of such paths using $(i, j)$ as the central segment is $(d_i - 1)(d_j - 1)$, where $d_i$ and $d_j$ are the degrees of $i$ and $j$, respectively. The total number of such paths is:

$$\text{\#L3\_path} = \sum_{(i,j) \in E} (d_i - 1)(d_j - 1) = \mathcal{O}\left(\sum_{(i,j) \in E} d_i d_j\right)$$

- **Computation per path.** For each length-3 path $u \to i \to j \to v$, CHA computes the iLCL and eLCL of intermediate nodes $i$ and $j$ with respect to the seed pair $(u, v)$.

  Each such computation, i.e., evaluating the iLCL/eLCL of node $i$ with respect to $(u, v)$, requires scanning the neighborhood of $i$ and takes $\mathcal{O}(d_i)$ time. However, this computation is performed only once for each triplet $(u, v, i)$, and the result is reused across all paths in which $(u, v, i)$ appears.

  Since each such triplet $(u, v, i)$ is associated with $\mathcal{O}(d_i)$ paths on average, the total cost is distributed across multiple paths. Thus, the amortized cost per path remains $\mathcal{O}(1)$.

- **Overall time complexity.** For compact notation, we define the RMS degree–degree product over edges:

$$\tilde{d}_{\text{edge}} = \sqrt{\frac{1}{m} \sum_{(i,j) \in E} d_i d_j}$$

and upper bound the total complexity as:

$$\mathcal{O}\left(m \cdot \tilde{d}_{\text{edge}}^2\right)$$

We now analyze this quantity under three typical network regimes:

- *Sparse, degree-homogeneous:* If the graph is sparse (i.e., $\bar{d} = \mathcal{O}(1)$) with relatively uniform degrees (i.e., $d_i = \mathcal{O}(1)$ for all nodes), then $d_i d_j = \mathcal{O}(1)$ for all edges and $m = \mathcal{O}(n)$. This yields:

$$\mathcal{O}\left(m \cdot \tilde{d}_{\text{edge}}^2\right) = \mathcal{O}(n)$$

So the overall time complexity of $\mathcal{O}(n)$.

- *Sparse, degree-heterogeneous:* If the graph is sparse (i.e., $\bar{d} = \mathcal{O}(1)$), but has a skewed degree distribution (e.g., power law), we upper bound:

$$\tilde{d}_{\text{edge}}^2 = \frac{1}{m} \sum_{(i,j) \in E} d_i d_j \leq \max_{(i,j)} d_i d_j$$

  In the worst case, this maximum can scale as $\mathcal{O}(n^2)$. Since the total number of edges is $m = \frac{n \cdot \bar{d}}{2}$, it follows that $m = \mathcal{O}(n)$. This leads to an overall complexity:

$$\mathcal{O}(m \cdot \tilde{d}_{\text{edge}}^2) = \mathcal{O}(n \cdot n^2) = \mathcal{O}(n^3)$$

- *Dense networks:* If the network is dense ($m = \mathcal{O}(n^2)$ and degrees are $\mathcal{O}(n)$), then $\tilde{d}_{\text{edge}} = \mathcal{O}(n)$ and:

$$\mathcal{O}(m \cdot \tilde{d}_{\text{edge}}^2) = \mathcal{O}(n^4)$$

**Summary.** The overall time complexity of CHA depends on the path length $\ell$ (only when $l \geq 4$) and the structural characteristics of the network. For both $\ell = 2$ and $\ell = 3$, we observe the following regimes:

- *Sparse, degree-homogeneous:* When the average degree is $\mathcal{O}(1)$ and degree distribution is uniform, the complexity is: $\mathcal{O}(n)$

- *Sparse, degree-heterogeneous:* When the average degree is $\mathcal{O}(1)$ but degree distribution is skewed (e.g., power-law), the complexity is higher due to hubs: $\mathcal{O}(n^3)$

- *Dense networks:* When the average degree is $\mathcal{O}(n)$, the worst-case complexity becomes $\mathcal{O}(n^4)$

Although CHA internally evaluates multiple CH models and path lengths, all models operate on the same set of intermediate-node statistics (iLCL and eLCL), which are computed only once. Therefore, evaluating multiple models does not increase the overall asymptotic complexity.

**Subranking Complexity.** The subranking step based on Spearman correlation requires computing all-pairs shortest paths, which has a worst-case time complexity of $\mathcal{O}(n^3)$.

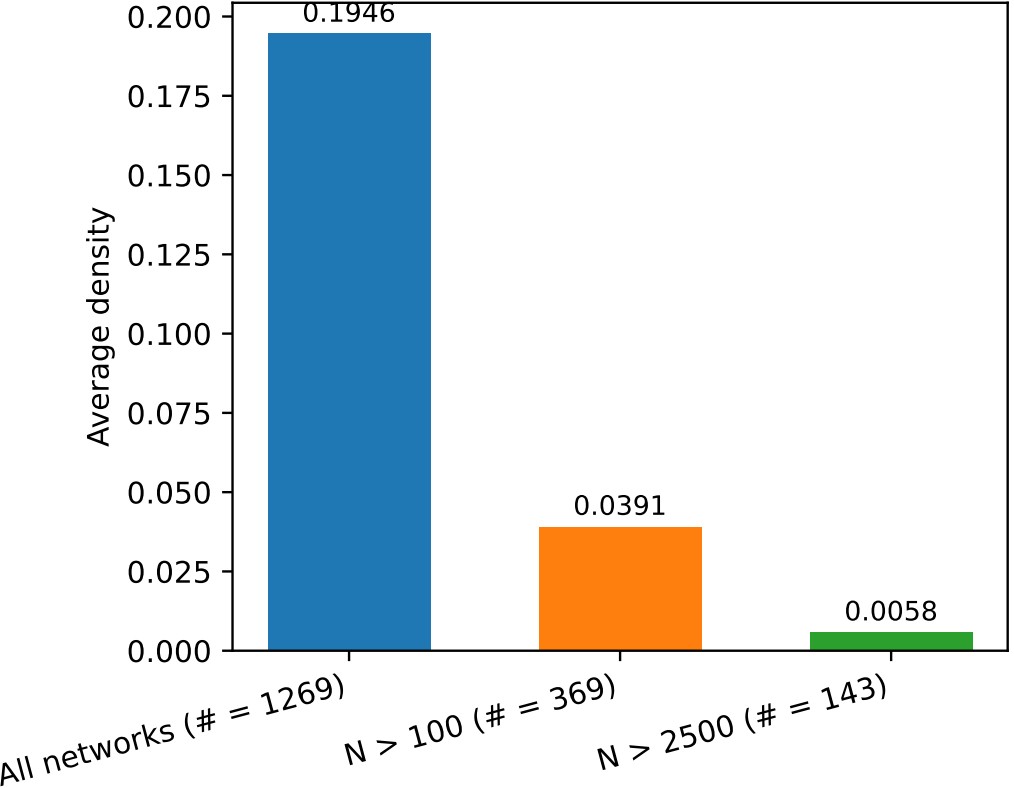

Figure 10: **Average edge density across different network sizes in ATLAS-static.** We report the average edge density, defined as the ratio between average degree and $N - 1$, for three subsets of the ATLAS-static dataset: all networks ($N = 1269$, density = 0.1946), networks with $N > 100$ ($n = 369$, density = 0.0391), and networks with $N > 2500$ ($n = 143$, density = 0.0058). The results show that large networks in ATLAS-static are extremely sparse, which makes CHA extremely fast to compute on such networks. (see Figure 11).

### E.2 Running Time of CHA

We empirically evaluate the running time of CHA under different graph conditions to validate its efficiency in practice.

Figure 10 reports the average edge density across networks of increasing size in the ATLAS-static dataset. While the overall dataset has moderately low density (0.1946), the average density for larger networks ($N > 2500$) drops sharply to 0.0058. This confirms that large real-world networks in ATLAS are typically sparse.

To further explore CHA's computational behavior under varying sparsity levels, Figure 11 shows its running time on synthetic networks with 500 nodes and increasing edge densities. The edge density is defined as the average degree divided by $N - 1$. As predicted by theoretical analysis, computation time increases steeply with density. For sparse graphs, CHA is significantly faster.

Together, these results indicate that CHA benefits from the inherent sparsity of real-world networks and achieves high scalability across the ATLAS benchmark.

### E.3 Time Complexity Comparison

We summarize here the computational complexity of CHA compared with major baselines, considering both *all-missing-links* and *subset* prediction settings.

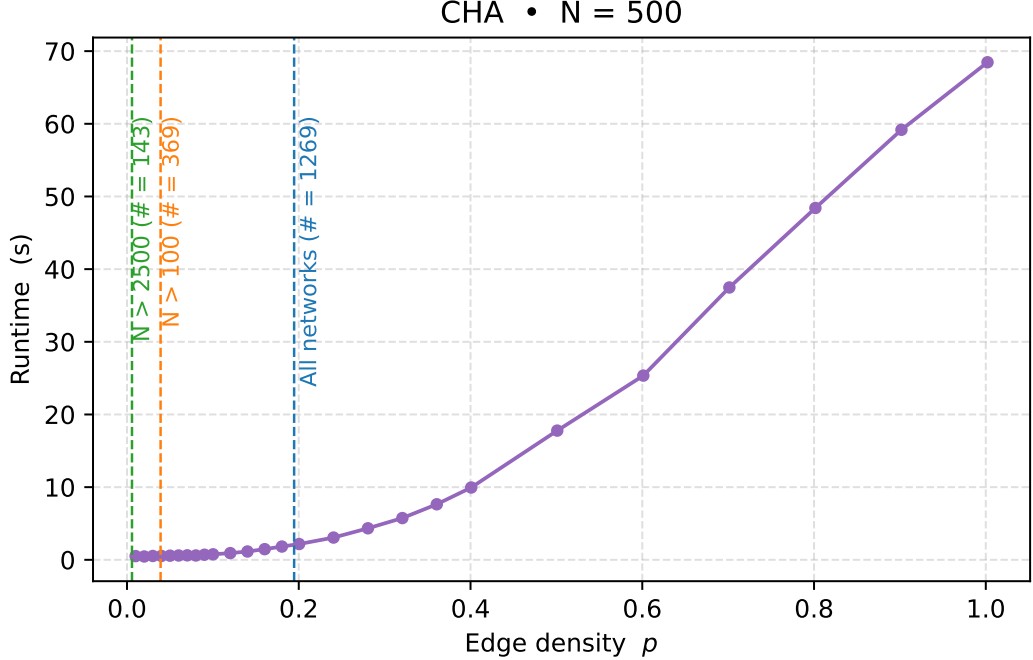

Figure 11: **Runtime of CHA on artificial networks with increasing density.** We generate synthetic networks with $N = 500$ nodes and vary the density, defined as average degree divided by $(N-1)$. The plot shows that CHA runs extremely fast on sparse networks, while its runtime increases rapidly as density grows. This highlights the efficiency of CHA in low-density regimes, which are common in many real-world networks.

**All-missing-links evaluation.**   In this setting, positives are ranked against all non-observed links $(O(n^2))$. CHA requires only one pass to compute its local community statistics (iLCL/eLCL) and reuse them across all CH variants, giving the following asymptotic costs:

Table 4: **Asymptotic time complexity of CHA and baselines for all-missing-links evaluation.** All costs refer to the sparse-network regime and include both training and scoring over all non-observed links.

| Method | Time Complexity |
|---|---|
| CHA (degree-homogeneous) | $O(n)$ |
| CHA (degree-heterogeneous) | $O(n^3)$ |
| SBM variants | $O(M(n + B^2))$ |
| SPM | $O(n^3)$ |
| Graph embedding methods | $O(K n^2)$ |
| Message passing methods | $O(F n^{2+r})$ |

Here, $r \geq 2$ is the hop count, $F$ the node-signature dimension, and $K$ the embedding dimension. Note that these complexities include **both training and scoring over all non-observed links** (for several methods, the **scoring step** dominates). $M$ is the number of missing links in the network, and $B$ represents the number of blocks in the SBM family of models. For SBM-based models, the primary computational cost arises from fitting the network using MCMC equilibration and the posterior edge probability estimation. The listed complexity corresponds to the sparse version; for denser networks, the complexity increases to $O(M(n^2 + B^2))$.

**Subset prediction.**   When only a subset of missing links is evaluated ($t \ll n^2$), the complexity of MPLP follows the same analysis as in the original paper, $O(td^r F)$, where $d$ is the maximum degree,

$r$ the hop count, and $F$ the node-signature dimension. For CHA, the per-link complexity can be derived explicitly as follows.

- **L2 paths.**
  (1) Compute the common-neighbor set $\mathcal{N}(u) \cap \mathcal{N}(v)$ in $O(d)$.
  (2) For each of the $O(d)$ common neighbors $z$, scan $\mathcal{N}(z)$ once to accumulate its iLCL/eLCL counts, each scan being $O(d)$.

  Total work per link: $O(d) \times O(d) = O(d^2)$.

- **L3 paths.**
  (1) Enumerate length-3 paths $u \rightarrow i \rightarrow j \rightarrow v$ by intersecting $\mathcal{N}(u)$ with the two-hop neighborhood of $v$, costing $O(d^2)$.
  (2) Each discovered path contributes one iLCL for $i$ and $j$; while enumerating, these counters are updated in place without an additional pass.

  Total work per link: still $O(d^2)$.

Therefore, CHA (without sub-ranking) runs in overall $O(td^2)$, which is asymptotically lower than MPLP+'s $O(td^r F)$ with $r = 2$ (as used in the MPLP article). Empirically, this theoretical difference is reflected in practice: on *ogbl-collab* (Table 5), CHA completes inference in about **6 seconds**, whereas MPLP+ requires roughly **6 minutes per run** on a single A100 GPU.

**Adaptive overhead.** Although CHA adaptively evaluates multiple CH variants and path lengths, all candidates reuse the same precomputed iLCL/eLCL statistics, computed once per graph, so the adaptive overhead is negligible in practice.

## F  Mapping Subranking to Likelihood Score

The CH sub-ranking mechanism refines the ordering of node pairs with tied CH scores by leveraging secondary scores (e.g., SPcorr). In some applications, it is desirable to map this refined ranking back to continuous likelihood scores.

To achieve this, we provide two interpolation-based strategies that preserve the original CH ranking while assigning distinct values to previously tied scores:

- **Score-guided interpolation.** Tied scores are adjusted based on the actual SPcorr values, preserving their relative magnitudes within the group. This results in a smooth, value-aware distribution of scores.
- **Rank-based interpolation.** Tied scores are redistributed uniformly according to their sub-rank positions, regardless of the SPcorr values. This maintains only the order but not the magnitude.

Both strategies are optional and not used during the main CHA evaluation, but are supported for downstream scenarios requiring continuous-valued outputs.

## G  Experiment on ogbl-collab

The *ogbl-collab* dataset [48] is a large-scale author collaboration graph with about 200K nodes, where each node represents an author and edges indicate co-authorships. The task is to predict future collaborations given the past, evaluated by **Hits@50**, ranking positive edges among randomly-sampled negative edges, following the standard metric used in previous link-prediction literature and the official OGB leaderboard protocol.

MPLP+ results are averaged over 10 runs, whereas CHA is deterministic and reported from a single run. For efficiency, we use MPLP+, the faster variant of MPLP, and run all models on a single A100 GPU. For fairness, CHA is evaluated without its sub-ranking step, mirroring the approximation trade-off used by MPLP+.

We also introduce **MPLP+L2** and **MPLP+L3**, which restrict structural feature to paths of length 2 and 3, respectively, corresponding to the homophilic and synergetic regimes described by Cannistraci-Hebb theory. To account for potential hyperparameter sensitivity, we conducted a small search on batch size $B \in \{8192, 16384, 32768\}$ and report the best configuration for each variant.

Table 5: **Results on ogbl-collab.** CHA outperforms all MPLP+ variants in accuracy while being much faster. Each MPLP+ run takes about **6 minutes** on a single A100 GPU, whereas CHA completes inference in approximately **6 seconds**.

| Method | Test Hits@50 (%) |
|---|---|
| MPLP+ (with feature) | $66.47 \pm 0.94$ |
| MPLP+ (without feature) | $64.84 \pm 1.21$ |
| MPLP+L2 (without feature) | $63.97 \pm 1.10$ |
| MPLP+L3 (without feature) | $65.01 \pm 0.43$ |
| CHA (CH3_L3) | **66.85** |

As shown in Table 5, CHA achieves higher accuracy than all MPLP+ variants, even surpasses MPLP+ using node features, despite relying solely on topology. Each MPLP+ run takes about **6 minutes** on a single A100 GPU, whereas CHA completes inference in approximately **6 seconds**. The superior performance of MPLP+L3 over MPLP+L2 further mirrors CHA's adaptive preference for L3-based (synergetic) rules, confirming the broader relevance of the Cannistraci-Hebb theory beyond CHA.

## H   Evaluation on Classical Non-Attributed Networks

We further evaluate CHA on the eight non-attributed networks used in the original MPLP paper [11]: USAir, NS, PB, Yeast, Celegans, Power, Router, and Ecoli. For a fair comparison, both MPLP and MPLP+ are run using the official hyperparameters provided for each network in the authors' repository. All results are reported under the all-missing-links evaluation protocol, where positives are ranked against all non-observed links rather than a 1:1 sampled subset. This protocol offers a more realistic assessment, since in real-world applications (e.g., protein–protein interactions) the true fraction of missing links is unknown, and 1:1 sampling tends to simplify the task artificially.

Table 6: **Comparison on the 8 additional non-attributed networks.** Evaluation metric is AUPR (higher is better).

| Network | MPLP+ | MPLP | CHA |
|---|---|---|---|
| USAir | 0.4948 | 0.4708 | 0.4795 |
| NS | 0.5259 | 0.3363 | 0.6823 |
| PB | 0.2093 | 0.1723 | 0.1984 |
| Yeast | 0.5105 | 0.4251 | 0.4652 |
| Celegans | 0.1315 | 0.1350 | 0.1285 |
| Power | 0.0069 | 0.0055 | 0.0092 |
| Router | 0.0673 | 0.0573 | 0.0960 |
| Ecoli | 0.5615 | 0.5603 | 0.5876 |
| Average | 0.3135 | 0.2703 | **0.3308** |

As shown in Table 6, CHA achieves the highest average AUPR, demonstrating better performance than both MPLP and MPLP+ across the classical non-attributed benchmarks.

## I   CH Theory for Prediction of Complex Network Connectivity

The Cannistraci-Hebb (CH) framework represents a *theory*, not a heuristic. Heuristics provide practical shortcuts without explanatory rigor, whereas a theory offers a systematic, empirically grounded framework capable of explaining and predicting phenomena. Over more than a decade of

studies, CH has demonstrated both predictive power and mechanistic interpretability in modeling link formation across complex systems. It unifies homophilic (similarity-driven) and synergetic (diversity-driven) principles of connectivity, forming a coherent explanation for self-organization in networks.

## J   On the Importance of Distinguishing Internal and External Connectivity in CH Theory

CH theory distinguishes between *internal* and *external* connectivity, describing how local communities emerge from the balance between intra- and inter-community links. Our experiments reveal that **external-link minimization alone** often achieves comparable or superior performance to full internal–external formulations. As shown in Figure 5, CH3 and CH3.1—based solely on external-link minimization—match or exceed CH2, suggesting that *"external-link minimization is all you need"* to drive the emergence of local-community structures essential for link prediction.

## K   Path-Length Extension Study

To examine whether longer paths provide additional predictive benefit, we extended CHA to evaluate path lengths L2–L6 on the ATLAS dataset. Table 7 reports the average AUPR win rates. Results show that performance peaks at L3, while longer paths (L4–L6) contribute negligibly, confirming that L2/L3 capture the essential topological scales for link prediction.

Table 7: **Average AUPR win rate of CHA with different path lengths on ATLAS.**

| Path length | Avg. AUPR win rate |
|---|---|
| L2 | 0.48 |
| L3 | **0.53** |
| L4 | 0.02 |
| L5 | 0.05 |
| L6 | 0.03 |

Accordingly, the main analysis focuses on L2 and L3, as they substantially outperform longer path configurations.

## L   Comparison with Variants of SBM

To assess the relationship between CHA and recent statistical models for missing-link prediction under the stochastic block model (SBM), we directly compare CHA with three SBM estimators [49]. We use the authors' official R implementation on the coauthorship network (Section 4.2.2 of the original paper). We then evaluate AUPR, NDCG, and Precision by ranking test positives among all missing links. As shown in Table 8, CHA substantially outperforms all SBM estimators across all metrics.

Table 8: **Comparison with SBM estimators from Gaucher & Klopp (2021)** on the coauthorship network. Evaluation metrics: AUPR, NDCG, and Precision (higher is better).

| Method | AUPR | NDCG | Precision |
|---|---|---|---|
| SBM-VAR | 0.1102 | 0.6910 | 0.2128 |
| SBM-missSBM | 0.1215 | 0.7046 | 0.2126 |
| SBM-softImpute | 0.1766 | 0.7030 | 0.2797 |
| CHA (CH3_L2) | **0.4567** | **0.8481** | **0.5073** |

