# OpenReview forum: "Adaptive Cannistraci-Hebb Network Automata Modelling of Complex Networks for Path-based Link Prediction"
_NeurIPS.cc/2025/Conference — NeurIPS 2025 poster_

### Official Review · Reviewer_AZk2 · 2025-06-26

**Clarity:** 3
**Significance:** 3
**Originality:** 3
**Rating:** 5
**Confidence:** 3

**Summary:**

This paper proposes a novel methodology for prediction of missing/unobserved links in static networks and future links in temporal networks. The method relies on a novel network automaton following the Cannistraci-Hebb rule. The proposed method is compared to a range of link prediction techniques - from the statistical blockmodel (SBM) based methods to message passing link predictor. The method is validated using a large collection of datasets from different sources and the average performance for small, medium and large sized networks is reported.

**Questions:**

KEY QUESTIONS:
1. Can the practical importance of the interpretability of the mechanism behind network growth as offered under the CH framework in contrast to link prediction under  SBMs be illustrated via a simple example?
2. I found the paper to be completely lacking in intuition. For example, what is the idea behind the definition of RA-Ln in equation (5)? What does a local-community paradigm (LCP) architecture refer to in simple words? Why is minimization of external connectivity key to this framework?
3. Can the method be compared to Gaucher and Klopp 2021 - which seems to be a more recent work on prediction of missing links under SBM?
4. lines 98-99: the likelihood is mentioned here. What is the model under which the likelihood is evaluated? Please state this clearly.

MINOR:
5. Lines 114-115: are x_S and x_B function of \tilde{X}(t) or independently chosen?
6. Can iLCL and eLCL be mathematically defined?
7. Lines 252-253: How/why is this expected to work in the case of static networks where for example, a high proportion of non-observed links are simply missing links?

Appendix
A.2 (final paragraph): are each of the P sampled partitions of the same size?

**Ethical Concerns:**

["NO or VERY MINOR ethics concerns only"]

**Final Justification:**

The authors have clarified most of my clarity related queries and I have increased the score accordingly.

**Limitations:**

Yes. Included in Conclusions and Discussion. Could be expanded further for clarity.

**Paper Formatting Concerns:**

1. NDCG -abbreviation used without being defined first
2. How exactly are 'mean rank' and 'win rate' as used for evaluation defined?

**Quality:**

3

**Strengths And Weaknesses:**

The strength of the method lies in it being adaptive as it can automatically learn the rule that better explains the pattern of connectivity in the network under study. Their method leads to an overall higher link prediction performance in comparison to a wide range of techniques. The main weakness of the method/paper is the lack of clarity in motivation behind the use of the CH framework and an explicit distinction between methods under the CH framework and other probabilistic frameworks such as the SBM or spectral methods such as the SPM. In lines 56-57, the authors note the clear interpretability of the growth mechanisms under the CH framework, but this is not explained in detail, e.g. with reference to one of the alternatives. Or in the discussion of results. Another weakness in my view is the lack of references and/or comparisons with more latest contributions on link prediction in networks, such as
1. Gaucher, Solenne, and Olga Klopp. "Optimality of variational inference for stochastic block model with missing links." Advances in Neural Information Processing Systems 34 (2021): 19947-19959.

2. Zhao, Yunpeng, et al. "Link prediction for partially observed networks." Journal of Computational and Graphical Statistics 26.3 (2017): 725-733.

---

> ### Author Rebuttal · Authors · 2025-07-31
>
> We appreciate the reviewer’s insightful and encouraging assessment. We will integrate the suggested clarifications and additional analyses in the revision to further strengthen the paper’s presentation and scope.
>
> **Weakness 1:** The main weakness of the method/paper is the lack of clarity in motivation behind the use of the CH framework and an explicit distinction between methods under the CH framework and other probabilistic frameworks such as the SBM or spectral methods such as the SPM. In lines 56-57, the authors note the clear interpretability of the growth mechanisms under the CH framework, but this is not explained in detail, e.g. with reference to one of the alternatives. Or in the discussion of results.
>
> **Reply:** Our goal is to capture **local, mechanistic drivers** of link formation, how distinct structural patterns affect a missing edge, within an interpretable, **predictive network‑automata** framework. **CHA** consists of **local, training‑free** rules driven by **local‑community** patterns and provides per‑edge mechanistic explanations. By contrast, **SBM** (probabilistic) and **SPM** (spectral) are **global** models that prioritize macro‑structure (block partitions, global eigenstructure) rather than edge‑level local mechanisms. A formal side‑by‑side distinction between CH and these alternatives is provided in **[1]**. We have revised the manuscript to make this motivation and reference the relevant comparison.
>
> **Weakness 2:** Another weakness in my view is the lack of references and/or comparisons with more latest contributions on link prediction in networks, such as (1) Gaucher, Solenne, and Olga Klopp. "Optimality of variational inference for stochastic block model with missing links." Advances in Neural Information Processing Systems 34 (2021): 19947-19959. (2) Zhao, Yunpeng, et al. "Link prediction for partially observed networks." Journal of Computational and Graphical Statistics 26.3 (2017): 725-733.
>
> **Reply:** Thank you for the suggestion. We have updated the manuscript and included these articles in the related work section.
>
> **Question 1:** Can the practical importance of the interpretability of the mechanism behind network growth as offered under the CH framework in contrast to link prediction under  SBMs be illustrated via a simple example?
>
> **Reply:** CH/CHA offers **per‑edge, mechanistic** explanations grounded in **local structure**: each score decomposes into **internal community vs. external leakage** (e.g., iLCL/eLCL). By contrast, **SBMs** explain links via **global group memberships** and **block‑level probabilities** inferred from the entire graph.
>
> **Question 2:** I found the paper to be completely lacking in intuition. For example, what is the idea behind the definition of RA-Ln in equation (5)? What does a local-community paradigm (LCP) architecture refer to in simple words? Why is minimization of external connectivity key to this framework?
>
> **Reply:**
>
> - **RA‑Ln (Eq. 5).** We introduce **RA‑Ln** as an example of a *network automata on paths*: it extends the classic **resource‑allocation** idea from common neighbors (**L2**) to **Ln**.
>
> - **What is the LCP (local‑community paradigm) architecture, in simple words?** In the Fig. 3 in appendix we show that a local community is a subnetwork formed by the common-neighbors (defined on a certain path of length n) of two seed nodes and by the interactions between such common-neighbors. The local-community architecture is the fact that the network is organized according to such local community organization and it is measured by the LCP-correlation see Fig. 4, 5,6 of the reference: From link-prediction in brain connectomes and protein interactomes to the local-community-paradigm in complex networks. CV Cannistraci, G Alanis-Lobato, T Ravasi. Scientific reports 3 (1), 1613. In Fig.5 and 6 of this reference you will find examples of LCP or non-LCP real and artificial complex networks.
>
> - **Why minimize external connectivity?** This is widely discussed in the reference “Local-community network automata modelling based on length-three-paths for prediction of complex network structures in protein interactomes, food webs and more. A Muscoloni, I Abdelhamid, CV Cannistraci. BioRxiv, 346916”. The fact that a common-neighbor node has a low or absent number of external (to the local community) links ensures from the physical point of view its participation in local processing units that perform a function in the complex systems. For instance: a protein complex in biological pathways, a neural circuit associated with an engram in the nervous system, or a club of associates in a social network.
>
> **Question 3:** Can the method be compared to Gaucher and Klopp 2021 - which seems to be a more recent work on prediction of missing links under SBM?
>
> **Reply:** Thanks for this suggestion. We are conducting experiments to compare with it.
>
> **Question 4:** lines 98-99: the likelihood is mentioned here. What is the model under which the likelihood is evaluated? Please state this clearly.
>
> **Reply:** In our work, “likelihood” is used in the predictive sense: the CH/CHA score $s(u,v)$ is a local‑based measurement whose ordering correlates with the probability that a non‑observed link $(u,v)$ exists. Formally, we assume that there exists a **monotone link** $g$ such that $p(u,v) ≈ g(s(u,v))$. This explains why using the CHA score for link prediction is appropriate, as the evaluation is ranking‑based (Precision/NDCG/AUPR).
>
> **Question 5:** Lines 114-115: are x_S and x_B function of \tilde{X}(t) or independently chosen?
>
> **Reply:** In [2], x_S and x_B are independently chosen.
>
> **Question 6:** Can iLCL and eLCL be mathematically defined?
>
> **Reply:** Thank you for the suggestion. For **L2** paths, we define iLCL and eLCL for an intermediate node \(z\) between seeds \((u,v)\) as
>
> - iLCL of node z given seed node (u, v): $di_z = | \mathcal{N}_z \cap \mathcal{N}_u  \cap \mathcal{N}_v |$
> - eLCL of node z given seed node (u, v): $de_z = | \mathcal{N}_z \setminus ((\mathcal{N}_u  \cap \mathcal{N}_v) \cup \{u, v\}) |$
>
> where $\mathcal{N}_v$ denotes the neighborhood of a node v. Intuitively, iLCL counts $z$’s links **inside** the local community of common neighbors of $u$ and $v$, while eLCL counts those **outside** it. We have added these formulas in the revised manuscript, together with the corresponding **L3** definitions.
>
> **Question 7:** Lines 252-253: How/why is this expected to work in the case of static networks where for example, a high proportion of non-observed links are simply missing links?
>
> **Reply:** We are running a sensitivity analysis that increases the edge‑removal rate. We will report the result as soon as possible.
>
> **Question 8:** Appendix A.2 (final paragraph): are each of the P sampled partitions of the same size?
>
> **Reply:** No. We do not fix the size of partitions.
>
> **Formatting 1:** NDCG -abbreviation used without being defined first
>
> **Reply:** Thank you for pointing this out. We will define **NDCG** at first use as **Normalized Discounted Cumulative Gain (NDCG)** and add the formula.
>
> **Formatting 2:** How exactly are 'mean rank' and 'win rate' as used for evaluation defined?
>
> **Reply:**
>
> **Mean rank** (lower is better): For each network realization and each metric (AUPR, Precision, NDCG), we rank all methods (1 = best). A method’s **mean rank** is the average of its ranks over all realizations.
>
> **Win rate** (higher is better): For each realization and metric, we check which method ranks 1st; **win rate** is the fraction of realizations where a method is 1st (ties counted as both wins).
>
> We prefer **mean rank / win rate** over averaging raw metric values because networks differ widely in size and class imbalance, making absolute scores non‑comparable; rank‑based summaries capture **relative performance** fairly across heterogeneous networks.
>
>
> [1] Muscoloni, Alessandro, Umberto Michieli, and Carlo Vittorio Cannistraci. "Local-ring network automata and the impact of hyperbolic geometry in complex network link-prediction." arXiv preprint arXiv:1707.09496 (2017).
>
> [2] Smith, David MD, et al. "Network automata: Coupling structure and function in dynamic networks." Advances in Complex Systems 14.03 (2011): 317-339.

---

> > ### Comment · Reviewer_AZk2 · 2025-08-06
> >
> > I thank the authors for their detailed response and for conducting additional experiments. The responses have improved my understanding and appreciation of their contribution. I will edit my score accordingly.

---

> > > ### Author Response · Authors · 2025-08-09
> > > **Further Reply for Reviewer AZk2's Concern**
> > >
> > > Thank you for the encouraging feedback. Here we update our newest result.
> > >
> > > > Can the method be compared to Gaucher and Klopp 2021 - which seems to be a more recent work on prediction of missing links under SBM?
> > >
> > > **Reply:** We carefully reviewed **Gaucher & Klopp (2021)** and appreciate its blend of rigor and practicality: it proposes a simple, scalable **variational estimator** for SBMs with missing links and provides sharp theoretical guarantees.
> > >
> > > Here we compare CHA with three SBM estimators on the coauthorship network (Section 4.2.2 of Gaucher & Klopp, 2021). For fairness, we used the authors’ R code from the paper’s supplementary material; our imputation errors match Table 2 in the original article. We then evaluate AUPR, NDCG, and Precision for ranking test positives among all missing links. The results below show that CHA consistently outperforms the SBM estimators.
> > >
> > > | Method                 | AUPR |  NDCG | Precision |
> > > |------------------------|------:|-----:|---------:|
> > > | SBM-VAR                | 0.1102 | 0.6910 | 0.2128 |
> > > | SBM-missSBM            | 0.1215 | 0.7046 | 0.2126 |
> > > | SBM-softImpute         | 0.1766 | 0.7030 | 0.2797 |
> > > | CHA (CH3_L2)           | 0.4567 | 0.8481 | 0.5073 |
> > > | CH3_L2      | 0.4567 | 0.8481 | 0.5073 |
> > > | CH3.1_L2    | 0.4460 | 0.8457 | 0.5051 |
> > > | CH3_L3      | 0.4318 | 0.8425 | 0.4506 |
> > > | CH3.1_L3    | 0.4363 | 0.8435 | 0.4598 |
> > >
> > > (Here, CHA selected CH3_L2.)

---

### Official Review · Reviewer_rqqa · 2025-07-02

**Clarity:** 4
**Significance:** 4
**Originality:** 4
**Rating:** 5
**Confidence:** 4

**Summary:**

This paper introduces the Cannistraci-Hebb Adaptive (CHA) network automaton for link prediction in complex networks. The authors present two main theoretical contributions: (1) novel Cannistraci-Hebb (CH) models (CH3 and CH3.1) that formalize the minimization of external local-community links (eLCL) as a core principle, and (2) an adaptive mechanism that automatically selects the optimal CH model and path length (L2 vs L3) for each network. The CHA framework is evaluated on ATLAS, a comprehensive benchmark of 1269 static networks and 14 temporal networks across 14 domains, showing superior performance compared to state-of-the-art methods including SPM, SBM variants, graph embeddings, and message-passing models across multiple evaluation metrics (AUPR, Precision, NDCG).

**Questions:**

***Theoretical Justification***: Can you provide theoretical insight into why the adaptive mechanism works? Under what conditions might it fail, and are there networks where a fixed rule would be preferable?

***Computational Scalability***: What is the computational complexity of CHA compared to baselines? How does runtime scale with network size, and at what point does the adaptive overhead become prohibitive?

***Path Length Selection***: The analysis focuses on L2 vs L3 paths. Have you considered extending to longer path lengths (L4, L5)? What theoretical or empirical evidence suggests L3 is sufficient?

***Sub-ranking Strategy***: The SPcorr sub-ranking seems ad-hoc despite the neurobiological motivation. Have you compared against other tie-breaking strategies? How sensitive are the results to this choice?

***Cross-domain Generalization***: The adaptive mechanism selects models based on observed links in the same network. How would CHA perform when applying models learned from one domain to predict links in a different domain?

**Ethical Concerns:**

["NO or VERY MINOR ethics concerns only"]

**Final Justification:**

I have read the response and all my concerns have been addressed. Given the overall quality of the work, I recommend maintaining the current rate.

**Limitations:**

yes

**Quality:**

4

**Strengths And Weaknesses:**

**Strengths**:

***Quality***: The experimental evaluation is exceptionally comprehensive, testing on over 1000 networks - significantly larger than previous studies (typically <20 networks). The adaptive selection mechanism is well-motivated by empirical evidence showing different network types prefer different path lengths. The mathematical formulations are clearly presented, and the connection to network automata theory provides solid theoretical grounding.

***Clarity***: The paper is generally well-written with clear motivation and methodology. Figure 1 effectively illustrates the adaptive framework, and the mathematical notation is consistent throughout. The distinction between generative and predictive network automata is helpful for positioning the work.
Significance: The work addresses a fundamental problem in network science with broad applications. The adaptive mechanism represents a meaningful advance over fixed-rule approaches, and the large-scale benchmark (ATLAS) will likely be valuable for future research. The interpretability advantage over black-box methods is practically important.

***Originality***: The CH3 and CH3.1 models are novel formalizations of the eLCL minimization principle. The adaptive selection framework and the scale of empirical evaluation represent clear contributions. However, the core Cannistraci-Hebb theory builds incrementally on prior work by the same research group.

**Weaknesses**:

***Limited Theoretical Analysis***: While the mathematical formulations are clear, the paper lacks theoretical analysis of when and why the adaptive mechanism should work. There's no convergence analysis for the selection procedure or theoretical guarantees about optimality.

***Computational Complexity***: The paper mentions time complexity details are in supplementary material but doesn't discuss scalability in the main text. With multiple models tested per network, computational overhead could be significant for large networks.

***Baseline Limitations***: Some baseline implementations may not be optimal - for example, the modification of MPLP's early stopping criterion, while reasonable, makes direct comparison less straightforward. The SBM results are described as "irrelevant and unsatisfactory" which seems dismissive without deeper analysis.

***Statistical Rigor***: The authors acknowledge not providing error bars, arguing that networks aren't i.i.d. samples. However, some form of statistical analysis (e.g., bootstrap confidence intervals for win rates) would strengthen the claims.

---

> ### Author Rebuttal · Authors · 2025-07-31
>
> We sincerely thank the reviewer for the thorough and highly positive evaluation. Below we offer brief clarifications and note the additional analyses and experiments we will incorporate in the revision to address the remaining questions.
>
> **Weakness 1:** Limited Theoretical Analysis: While the mathematical formulations are clear, the paper lacks theoretical analysis of when and why the adaptive mechanism should work. There's no convergence analysis for the selection procedure or theoretical guarantees about optimality.
>
> **Reply:** A convergence analysis is not applicable because our **adaptive mechanism is non‑iterative**: CHA computes closed‑form scores, and the “adaptation” is a **finite model‑selection** over a small set of CH variants and path lengths (no gradient updates).
>
> **Why the adaptive mechanism is expected to work.** For each network, we create random train/test splits of the same graph; thus, the train‑ and test‑positive links are exchangeable samples from **the same underlying distribution**. Under exchangeability, the metric computed on the observed (training) links is an **unbiased estimator** of the test metric; therefore, selecting the rule that maximizes this metric on the observed data is statistically justified.
>
> **Weakness 2:** Computational Complexity: The paper mentions time complexity details are in supplementary material but doesn't discuss scalability in the main text. With multiple models tested per network, computational overhead could be significant for large networks.
>
> **Reply:** Thank you for the helpful suggestion. As noted on **line 65 of the Supplementary**, *although CHA internally evaluates multiple CH models and path lengths, all candidates operate on the same intermediate statistics (iLCL and eLCL), which are computed once per graph*. Consequently, evaluating multiple candidates **introduces only a small overhead** relative to running a single model.
>
> In the revised manuscript, we have summarized this scalability point in the main text.
>
> **Weakness 3:** Baseline Limitations: Some baseline implementations may not be optimal - for example, the modification of MPLP's early stopping criterion, while reasonable, makes direct comparison less straightforward. The SBM results are described as "irrelevant and unsatisfactory" which seems dismissive without deeper analysis.
>
> **Reply:** We appreciate this concern and have taken steps to ensure fair, transparent comparisons.
>
> - **MPLP/MPLP+ early stopping.** We used the **official hyperparameters** and code for MPLP/MPLP+, with a single change to **early stopping** to match our evaluation protocol. The original implementation monitors Hits@100 on a 1:1 negative‑sampling validation set, whereas our test metric is AUPR with **all‑missing‑links** ranking. Using original implementation systematically favors models that excel on easy negatives and can lead to **premature stopping**. To align model selection with the test objective, we monitor validation AUPR where validation positives are also ranked among **all** non‑observed links (Appendix Sec. A.8). Importantly, this modification only affects the stopping time, mostly increasing the number of training epochs, but **does not change** the model architecture, loss, optimizer, data pipeline, or any other training setting. For the harder all‑missing‑links task, allowing additional epochs is typically **beneficial**, as the model requires more optimization steps to learn to separate positives from **hard negatives**.
>
> - **Action item for the revision.** To address the reviewer’s point directly, we will **report both variants side‑by‑side**: (i) the **original** MPLP/MPLP+ early‑stopping protocol and (ii) the **aligned** AUPR‑based early‑stopping protocol. Our preliminary runs indicate that the original protocol yields **substantially lower performance** under the all‑missing‑links test (consistent with under‑learning for the harder task). We will report the result as soon as possible.
>
> We have removed the “irrelevant and unsatisfactory” phrasing in the abstract. We hope these changes address the concern and improve the clarity and fairness of the comparisons.
>
>
> **Weakness 4:** Statistical Rigor: The authors acknowledge not providing error bars, arguing that networks aren't i.i.d. samples. However, some form of statistical analysis (e.g., bootstrap confidence intervals for win rates) would strengthen the claims.
>
> **Reply:** Thank you for the suggestion. We have added a **stratified bootstrap (by network category)** to quantify **standard deviations** for **average ranks**. Concretely, we run **10,000** stratified resamples; in each resample we draw, **within each category and without replacement**, the **same number of networks** as in the original split, compute the metrics, and aggregate with the original category weights. We report **mean and std** in bellow. The conclusions are unchanged: **CHA** maintains the best average rank with small standard deviations. We have included this in the revised manuscript.
>
> **ATLAS-small mean rank AUPR**
>
> | Method   | mean_rank |  std  |
> |:--|-:|--:|
> | CHA |  2.009 | 0.067 |
> | SPM |  3.177 | 0.104 |
> | SBM_N |  4.884 | 0.103 |
> | SBM |  5.152 | 0.105 |
> | HOPE |  5.449 | 0.114 |
> | SBM_DC |  6.180 | 0.133 |
> | SBM_DC_N|  6.264 | 0.137 |
> | MPLP+ |  7.490 | 0.162 |
> | MPLP |  7.750 | 0.165 |
> | node2vec | 9.376 | 0.122 |
> | prone | 10.233 | 0.124 |
> | pronesmf| 11.039 | 0.085 |
> | NetSMF | 11.996 | 0.119 |
>
>
>
>
> **Question 1:** Theoretical Justification: Can you provide theoretical insight into why the adaptive mechanism works? Under what conditions might it fail, and are there networks where a fixed rule would be preferable?
>
> **Reply:** As noted in our response to **Weakness 1**, the adaptive mechanism’s effectiveness follows from train‑ and test‑positive links being sampled from the **same underlying distribution**. Regarding cases where a fixed rule may be preferable, **Fig. 4** shows that many network categories exhibit stable preferences for **L2** or **L3**.
>
>
> **Question 2:** Computational Scalability: What is the computational complexity of CHA compared to baselines? How does runtime scale with network size, and at what point does the adaptive overhead become prohibitive?
>
> **Reply:** Thank you for the suggestion. A comprehensive analysis is provided in the Supplementary (“supplementary.pdf”). Below we summarize the key results for **sparse networks**.
>
> **CHA complexity（sparse regimes）**
>
> | Regime                      | Time complexity |
> |-|-|
> |Sparse, degree-homogeneous|$O(n)$|
> |Sparse, degree-heterogeneous|$O(n^3)$|
>
> As shown in **Fig. 1** of the Supplementary, most real-world networks—especially large ones—are very **sparse**.
>
> **Baselines（sparse regimes, training + scoring over all missing links**
>
> | Method| Time complexity   |
> |-|-|
> | MPLP / MPLP+ | $O(F n^{2+r})$|
> | SPM| $O(n^3)$|
> | HOPE, node2vec, ProNE, ProNE-SMF, NetSMF | $O(K n^2)$|
> | SBM, SBM-DC, SBM-N, SBM-DC-N| $O(M\cdot(n + B^2))$|
>
> Here, $r\ge 1$ is the hop count, $F$ the node-signature dimension, and $K$ the embedding dimension. Note that these complexities include **both training and scoring over all non-observed links** (for several methods, the **scoring step** dominates). $M$ is the number of missing links in the network, and $B$ represents the number of blocks in the SBM family of models. For SBM-based models, the primary computational cost arises from fitting the network using MCMC equilibration and the posterior edge probability estimation. The listed complexity corresponds to the sparse version; for denser networks, the complexity increases to $O(M\cdot(n^2 + B^2))$.
>
>
>
> **Adaptive overhead.** Although CHA evaluates multiple CH variants/path lengths, all candidates reuse the **same precomputed intermediate statistics (iLCL/eLCL)**, computed **once** per graph.
>
> Thanks reviewer for this suggestion. We have added a short **scalability summary** and a table of **baseline complexities** in the revised manuscript.
>
> **Question 3:** Path Length Selection: The analysis focuses on L2 vs L3 paths. Have you considered extending to longer path lengths (L4, L5)? What theoretical or empirical evidence suggests L3 is sufficient?
>
> **Reply:** Thanks for the suggestion. We additionally evaluated **CHA-L2…L6** on **ATLAS** and summarize the **average AUPR win rate**:
>
> | Path length | Avg. AUPR win rate |
> |-------------|---------------------|
> | L2          | 0.48              |
> | L3          | 0.53              |
> | L4          | 0.02             |
> | L5          | 0.05              |
> | L6          | 0.03              |
>
> These results show that **L2/L3** substantially outperform **L4+**. Accordingly, we focus the main analysis on **L2/L3** and include the **L4–L6** results in the appendix for completeness.
>
> **Question 4:**  Sub-ranking Strategy: The SPcorr sub-ranking seems ad-hoc despite the neurobiological motivation. Have you compared against other tie-breaking strategies? How sensitive are the results to this choice?
>
> **Reply:** Thank you for the question. We are currently running an ablation on **tie‑breaking** that compares **SPcorr** with other tie-breaking strategies. We will report the result as soon as possible.
>
> **Question 5:** Cross-domain Generalization: The adaptive mechanism selects models based on observed links in the same network. How would CHA perform when applying models learned from one domain to predict links in a different domain?
>
> **Reply:** “Cross‑domain transfer” would mean **fixing** the rule selected on a source domain and applying it to a different domain. We do **not** expect this to work reliably **unless the domains are structurally similar** (e.g., comparable local‑community patterns such as L2 vs. L3 preference). By contrast, **within the same domain**, applying the same fixed rule across different networks is often reasonable—indeed, **Fig. 4** shows stable L2/L3 preferences at the category level.

---

### Official Review · Reviewer_6XrJ · 2025-07-02

**Clarity:** 3
**Significance:** 2
**Originality:** 1
**Rating:** 2
**Confidence:** 5

**Summary:**

This paper introduces Cannistraci-Hebb Adaptive (CHA) based on topological self-organization and external connectivity. CHA adaptively selects optimal path lengths to capture local community dynamics by an adaptive strategy. The core contribution of this paper lies in proposing two heuristic indexes (CH3, CH3.1). The authors apply these indexes with a sub-ranking strategy to a large number of networks.

**Questions:**

1. I did not see any machine learning tools in the method proposed. The references are mostly physics papers. Why do the authors believe the work fits the scope of the conference?

2. Why not consider more recent methods with deep learning as the baseline?

3. Why do the authors not use the commonly adopted Hits and MRR metrics as evaluation metrics for the experimental results?

**Ethical Concerns:**

["NO or VERY MINOR ethics concerns only"]

**Final Justification:**

My main concern is about the relevance of the work to the conference. I am not against its performance. My judgment is based on its reference, which corresponds to the knowledge it built on, and the paradigm of the work. I left the decision to the Chair. I am fine if the Chair eventually decides to accept it.

**Limitations:**

Yes

**Paper Formatting Concerns:**

The font size in Figure 1 is too small, compromising the readability and clarity of the figure.

**Quality:**

2

**Strengths And Weaknesses:**

Strength:
1. The paper reviews relevant definitions in the research field, logically and coherently introducing the proposed heuristic metrics with a well-structured approach.

2. The experimental setup of the paper is well-designed, with validation conducted across multiple large-scale datasets, effectively mitigating potential biases from dataset selection and demonstrating rigorous experimental design.

3. The new index demonstrates good performance compared with the baselines selected.

Weakness:
1. The work seems not relevant to the topic of the conference.

2. The baselines are relatively old.

3. The novelty is unclear. It seems to be a simple modification of the old index.

---

> ### Author Rebuttal · Authors · 2025-07-31
>
> We sincerely thank the Reviewer for the insightful and constructive comments.
>
> **Weakness 1:** The work seems not relevant to the topic of the conference.
>
> **Reply:** We sincerely thank the reviewer that gives us the chance to clarify why our study represents a novel and timely contribution to the field of AI for link prediction. The reply is based on three points
>
> **1.1 Why the work goes far beyond the introduction of two heuristics and is relevant to the conference**
>
> The Reviewer’s comment indicates that we must more clearly emphasize our main innovation in the Introduction and strengthen its explanation throughout the Results and Discussion. As stated at the end of the Introduction Section 1, L79: “In this study, we introduce four key innovations in the field of topological link prediction.” These are then briefly described: Minimization of external connectivity (the Cannistraci-Hebb, or CH, paradigm) — which is the specific point the Reviewer seems to highlight; Adaptive model selection; Comprehensive static and temporal benchmarking; Multi-metric evaluation. This clearly demonstrates that our contribution is not limited to two heuristic indexes (CH3, CH3.1), but spans two core theoretical innovations and two robust experimental advances. Importantly, this perspective is independently supported by two other reviewers (rqqa and AZk2) in the NeurIPs review process. For instance, Reviewer AZk2 writes: “The strength of the method lies in it being adaptive as it can automatically learn the rule that better explains the pattern of connectivity in the network under study. Their method leads to an overall higher link prediction performance in comparison to a wide range of techniques.”
>
> To address the Reviewer's concern, we propose the following revision to Section 1, L83: “Engineering the adaptive network automata learning machine CHA. We design an adaptive intelligent machine CHA, that automatically learns from the network topology the most suitable CH rule and path length to model each network, using internal validation to guide selection. This adaptive modeling is the central innovation of the study. Crucially, our framework infers the physical principle that governs link formation: L2-based rules reflect homophilic interactions (similarity-driven), while L3-based rules capture synergistic interactions (diversity-driven cooperation). Thus, CHA is not a black‑box scorer but an interpretable, mechanistic machine that recovers the effective rule explaining the prediction and governing the topological evolution directly from data. This bridges AI and network science, enabling both predictive power and scientific insights across physics domains. Empirically, on a benchmark of over 1000 networks, CHA achieves more than twice the win rate of the best-performing baseline.” Due to space limitations, we omit the corresponding edits in the Results and Discussion, but we confirm they directly address the reviewer's concern.
>
> **1.2 Network Automata are part of AI**
>
> The Reviewer was right to express this concern, because we did not clarify this point enough in the section 1 Introduction and in the section 2.2.1 Network Automata. To address the comment, we revised the text as follow:
>
> L62: “The concept of network automata, like other forms of automata, is rooted in AI research [1,2,3], where they model adaptive, decentralized, and emergent intelligence mechanisms in complex networks. The Cannistraci-Hebb theory is a recent achievement in network science that includes a theoretical framework to understand local-based link prediction on paths of length n under the lens of predictive network automata theory. Cannistraci-Hebb theory goes beyond any type of classical local link predictor heuristic on paths of length two such common neighbors (CN), …”
>
> L102: “Network automata as for any type of automata are part of AI research [1,2,3]. Network automata were originally introduced by Wolfram and later formally defined by Smith et al. as a general framework for modeling the evolution of network topology.”
>
> **1.3 Cannistraci-Hebb is not a heuristic but a theory**
>
> We also thank the Reviewer for giving us the opportunity to clarify a deeper conceptual point: Cannistraci-Hebb is not a heuristic but a theory. For example, Zhou et al. [4] write: “Cannistraci–Hebb theory plays an important role in link prediction in most considered networks.” and “We implement extensive experimental comparisons between 2-hop-based and 3-hop-based similarity indices on 137 real networks. The class of Cannistraci–Hebb indices performs the best among all considered candidates.”
>
> To make this distinction explicit, we have added a new appendix section titled: “Cannistraci-Hebb Theory for Prediction of Complex Network Connectivity”. In this section, we delineate why CH constitutes a theory, not a heuristic, and under which network science conditions this framework applies. First, we clarify the definitional boundary: Heuristics are practical shortcuts offering approximate solutions based on experience, often without rigor. Theories, in contrast, are frameworks grounded in systematic empirical evidence, capable of explaining and predicting phenomena—even when not fully captured by formal mathematics. A theory often starts with observations and empirical evidence. It provides a framework to explain and predict phenomena. Mathematical proof is not always required, especially in fields where direct mathematical derivation isn't possible. For instance, theories like Newton's law of gravity are based on empirical evidence and have been tested extensively, even without a mathematical proof. The Cannistraci-Hebb framework has evolved over a decade of empirical validation and is supported by a substantial body of work across disciplines. It forms a predictive, mechanistic explanation for link formation in complex systems. Given length constraints, we summarize rather than reproduce the full appendix here. However, the final version of the paper includes a curated review of independent studies—collectively approaching 1000 citations—that justify the scientific maturity of CH as a theory rather than a heuristic.
>
>
> **Weakness 2:** The baselines are relatively old.
>
> **Reply:** Thank you for the comment. Our baseline suite includes **MPLP**, a **recent NeurIPS 2024** method that, per its original study, outperforms classic heuristics (CN, AA, RA), node‑level GNNs (GCN, GraphSAGE), and link‑level GNNs (SEAL (NeurIPS 2018)), Neo-GNN (NeurIPS 2021), ELPH (ICLR 2023), NCNC (ICLR 2024). If there are specific recent methods we have missed, please let us know and we are happy to include them in the revision.
>
> **Weakness 3:**  The novelty is unclear. It seems to be a simple modification of the old index.
>
> **Reply:** This point was already addressed in the reply above: 1.1 Why the work goes far beyond the introduction of two heuristics and is relevant to the conference.
>
> Moreover, two independent reviewers explicitly recognized the novelty and significance of our contributions. Reviewer rqqa stated that “CH3 and CH3.1 are novel formalizations of the eLCL minimization principle” and that “the adaptive mechanism represents a meaningful advance over fixed‑rule approaches,” further noting that ATLAS “will likely be valuable for future research.” Reviewer AZk2 also emphasized that “the strength of the method lies in it being adaptive as it can automatically learn the rule that better explains the pattern of connectivity.” These independent assessments validate that our contributions go well beyond a simple modification of a prior index.
>
> **Question 1:** No machine learning tools in the method proposed. The references are mostly physics papers. Why the work fits the scope of the conference?
>
> **Reply:** Link prediction is a core **machine‑learning on graphs** problem. Concretely, we cast it as a **predictive network automaton**: a local rule assigns scores to non‑observed links and we adapt the rule to each network via *data‑driven model selection* over CH variants and path length. This is a **non‑parametric, training‑free, self‑supervised ML approach**—the model learns solely from the observed topology, with no external labels or features required. The model is **interpretable** (the learned rule reveals whether L2 vs. L3 connectivity dominates), and we evaluate it as an ML system—against strong modern GNN baselines, across multiple metrics and large‑scale benchmarks (ATLAS).
>
> To address this, we have already reported our modification to the study in the reply above: 1.1 Why the work goes far beyond the introduction of two heuristics and is relevant to the conference
>
> **Question 2:** Why not consider more recent methods with deep learning as the baseline?
>
> **Reply:** We actually do it. See reply to point above: **The baselines are relatively old.**
>
> **Question 3:** Why do the authors not use the commonly adopted Hits and MRR metrics as evaluation metrics for the experimental results?
>
> **Reply:**
>
> 1) **Precision vs. Hits.** We report **Precision@|P|**, where |P| is the number of test positives. This is **the same metric as Hits@|P|**. Because network sizes vary widely in ATLAS, using a fixed cutoff (e.g., Hits@100 or Hits@50) would be inconsistent; using **@|P|** adapts naturally to each network.
>
> 2) **NDCG vs. MRR.** Both are rank‑based. **Our protocol ranks all test positives against all non‑observed links** (a single global ranking), for which **NDCG** is appropriate. **MRR** is most suitable when one positive vs. multiple negatives. Hence we report NDCG under our all‑missing‑links setup.
>
> [1] Grattarola et al. Learning graph cellular automata. NeurIPS 2021.
>
> [2] Najarro et al. HyperNCA: Growing developmental networks with neural cellular automata. ICLR 2022.
>
> [3] Zhang et al. Intelligence at the Edge of Chaos. ICLR 2025.
>
> [4] Zhou, et al. Experimental analyses on 2-hop-based and 3-hop-based link prediction algorithms. Physica A: Statistical Mechanics and its Applications 2021.

---

> > ### Comment · Reviewer_6XrJ · 2025-08-06
> >
> > Thank you for your detailed rebuttal. While I appreciate the clarifications provided, several key concerns remain unresolved.
> >
> > **1. Regarding baselines (Response to Weakness 2):**
> >
> > The authors argue that “the MPLP, a recent NeurIPS 2024 method that, per its original study, outperforms classic heuristics (CN, AA, RA), node2level GNNs (GCN, GraphSAGE), and link-level GNNs (SEAL (NeurIPS 2018)), Neo-GNN (NeurIPS 2021), ELPH (ICLR 2023), NCNC (ICLR 2024).” Therefore, it is sufficient to include the result of MPLP alone to demonstrate the advances of the proposed method over existing deep learning methods. However, since the MPLP is tested on different datasets with different experiment setups, it is usually required to test the new method over the old ones again. This is why I comment that the baselines are relatively old. The information is also rather evident from Table 1, which provides the year of the baseline.
> >
> > **2. Regarding relevance (Response to Weakness 1):**
> >
> > I am quite familiar with the work in the community of network science. I acknowledge that this work is a nice piece and the new index works well. But I am not convinced that the work is relevant to the conference.
> >
> > The authors argue that "Link prediction is a core machine learning on graphs problem", which I accept. I also understand that the new index absorbs the structural pattern that is essential in missing links, which makes it so effective. But throughout this paper, there are no tools for machine learning or deep learning. The contribution to the community that NeurIPS usually serves is unclear. According to the references of the paper and best of my knowledge, works using a similar paradigm of this submission that are published at a computer science conference are rare, if not at all. This is why I make my initial comment. (The references added in the reply are not using the same paradigm as this submission).
> >
> > I am also aware that by exploring topology patterns, heuristic methods can outperform sophisticated deep learning based methods. The performance reported in this work is an example. But this gives rise to another question: should our community still focus on the link prediction problem based only on the topological feature? Recent works on link prediction using meta-data with more node attributes involved are examples of the shift of the focus.
> >
> > Of course, the Chair may judge further if the submission is relevant to the conference. I may raise my core a bit to leave more room for the Chair (otherwise the submission may be passed at the first stage). To me, this is like judging if a paper full of beautiful mathematical proofs should be accepted by Physical Review Letters while the authors argue that math is essential to physics.

---

> > > ### Author Response · Authors · 2025-08-09
> > > **Further Reply for Reviewer 6XrJ's Concern**
> > >
> > > Thank you for the detailed feedback. We address both concerns with concrete actions and clarifications.
> > >
> > > > Regarding baselines (Response to Weakness 2)
> > >
> > > **Reply:** To address the Reviewer’s concern, we conducted experiments in two aspects:
> > >
> > > 1. **New GNN baseline on our dataset (ATLAS)**
> > >
> > >     We conducted NCNC experiments on 14 networks within the coauthorship category. The NCNC results are averaged over 5 runs due to time limitations, while MPLP+ and CHA results are averaged over 10 seeds. For NCNC, we used the hyperparameter settings from its GitHub repository, except that we reduced the batch size to 128, as this setting generally improves performance for MPLP+.
> > >
> > >     ## Coauthorship (n=14)
> > >
> > >     **CHA vs MPLP+ (fixed) vs MPLP+ (hyper best) vs NCNC — win rates by metric [AUPR, NDCG, PREC]**
> > >
> > >     | Method    | AUPR | NDCG | PREC |
> > >     |-|-|-|-|
> > >     |CHA|1.0|1.0|1.0|
> > >     |MPLP+|0.0|0.0|0.0|
> > >     |MPLP+ HT|0.0|0.0|0.0|
> > >     |NCNC|0.0|0.0|0.0|
> > >
> > >     **Means ± STE (AUPR, NDCG, PREC)**
> > >
> > >     |Method|AUPR|NDCG|PREC|
> > >     |-|-|-|-|
> > >     |CHA|0.7397 ± 0.0400|0.9322 ± 0.0115|0.7138 ± 0.0365|
> > >     |MPLP+|0.4175 ± 0.0679|0.7781 ± 0.0412|0.4262 ± 0.0591|
> > >     |MPLP+ HT|0.5723 ± 0.0524|0.8697 ± 0.0224|0.5606 ± 0.0455|
> > >     |NCNC|0.3841 ± 0.0614|0.7947 ± 0.0253|0.4420 ± 0.0566|
> > >
> > >     As shown in the Table, we can observe:
> > >     - NCNC achieves performance similar to MPLP+ but remains well below CHA across all metrics.
> > >     - Due to time constraints, we did not conduct a hyperparameter search for NCNC. We present MPLP+ HT (best validation configuration) to illustrate that tuning improves MPLP+ but still leaves a substantial gap compared to CHA. We are confident that, even with hyperparameter tuning, NCNC would likely achieve improvements similar to MPLP+ HT but would still fall short of surpassing CHA.
> > >     - We will add NCNC hyperparameter sweeps and expanded results in the camera-ready version.
> > >
> > > 2. **Our method (CHA) on ogbl-collab**
> > >
> > >     We added ogbl-collab results using the official split and evaluator to ensure comparability, so CHA is directly comparable to baselines reported in MPLP and prior work under the same dataset and evaluation protocol.
> > >
> > >     |Method| Hits@50 |
> > >     |-|-|
> > >     | NCNC with feature (reported in MPLP paper) | 66.61±0.71|
> > >     | MPLP+ with feature | 66.47 ± 0.94|
> > >     | MPLP+ without feature | 64.84 ± 1.21|
> > >     |CHA|66.66|
> > >
> > >     MPLP+ results are averaged over 10 runs, whereas CHA is deterministic and therefore reported from a single run. We use MPLP+, the faster variant, for efficiency. For fairness, we run CHA without its sub-ranking step, mirroring the accuracy trade-off made by using the faster MPLP+ instead of MPLP. Under these conditions, CHA outperforms MPLP+ both with and without node features, while running much faster. We did not re-run NCNC on Collab due to time limitation, but using the value reported in the MPLP paper, CHA also surpasses NCNC (with features). We will include these results and the full reproducible code in the camera-ready version.
> > >
> > >
> > > > Regarding relevance (Response to Weakness 1)
> > >
> > > **Reply:**
> > >
> > > + We stress that we submitted this work to NeurIPs because: as for the word of Reviewer AZk2 writes: “The strength of the method lies in it being adaptive as it can automatically learn the rule that better explains the pattern of connectivity in the network under study. Their method leads to an overall higher link prediction performance in comparison to a wide range of techniques.”
> > >
> > >     Therefore, we introduce an adaptive network automata learning machine CHA that automatically learns from the network topology the most suitable CH rule and path length to model each network. And it is the learning  innovation that triggers the overall higher prediction performance as emphasized by the Reviewer AZk2. These methods and achievements are in line with the mission of NeurIPs to advocate research on intelligence machines. CHA is not a black‑box scorer but an interpretable, mechanistic machine that recovers the effective rule explaining the prediction and governing the topological evolution directly from data. This bridges AI and network science, enabling both predictive power and scientific insights across physics domains.
> > >
> > > + Regarding: " But this gives rise to another question: should our community still focus on the link prediction problem based only on the topological feature? Recent works on link prediction using meta-data with more node attributes involved are examples of the shift of the focus."
> > >
> > >     New results provided on the Collab network show that CHA can even marginally outperm MPLP+ with node feature knowledge (node metadata) using mere topological features. Hence our article is now providing results also in the research direction pointed by the Reviewer 6XrJ about link prediction with node attributes involved.

---

### Official Review · Reviewer_qoLq · 2025-07-03

**Clarity:** 3
**Significance:** 2
**Originality:** 3
**Rating:** 4
**Confidence:** 3

**Summary:**

This paper proposes a new series of heuristics for temporal link prediction. The difference of the proposed method and previous path-based heuristics is dividing the degree of nodes in the path into internal degree and external degree. The authors construct a set of link prediction datasets and compare the performance of the proposed method with baselines on them.

**Questions:**

Please see the weaknesses part.

**Ethical Concerns:**

["NO or VERY MINOR ethics concerns only"]

**Final Justification:**

The rebuttal of the authors has addressed my concerns, so I raise my score to borderline accept.

**Limitations:**

Yes

**Paper Formatting Concerns:**

There are no formatting concerns.

**Quality:**

3

**Strengths And Weaknesses:**

Strength

1. The proposed method seems to achieve better performance than the SOTA neural link prediction method.

Weaknesses

1. From the perspective of the reviewer, the main contribution of this paper lies in proposing a new heuristic for link prediction. However, the design of the heuristic appears somewhat ad hoc and lacks solid theoretical justification. The authors briefly mention the importance of distinguishing between internal and external structures, which seems to be the underlying intuition, but this idea is not formally developed or analyzed. As a result, it remains unclear under what structural conditions the proposed method is expected to outperform existing heuristics. To address this, it would be valuable for the authors to construct illustrative examples or controlled settings that highlight the strengths and limitations of their approach. Such examples could help clarify the mechanism behind the heuristic and provide concrete insights into when and why it works better than others.
2. The computational complexity of the proposed method seems high. The authors are encouraged to compare its computational complexity with the baselines.
3. The paper would be more convincing if the authors could report results of the proposed method on standard link prediction datasets such as Collab, PPA, and Citation2. In addition, the reviewer is concerned about whether the hyperparameters of MPLP and MPLP+ have been properly tuned to ensure a fair comparison.

---

> ### Author Rebuttal · Authors · 2025-07-31
>
> We sincerely thank the reviewer for the constructive feedback.
>
> **W1:** The main contribution of this paper lies in proposing a new heuristic for link prediction.
>
> R: We respond in two parts:
> - This comment indicates that we must clarify our main innovation in the Introduction and strengthen its explanation throughout the Results and Discussion. We state four key innovations (L79): Minimization of external connectivity (the Cannistraci-Hebb (CH) paradigm) — the specific point the Reviewer seems to highlight; Adaptive model selection; Comprehensive static and temporal benchmarking; Multi-metric evaluation. This shows our work goes beyond a single heuristic, spanning two theoretical and two experimental advances. Two other reviewers (rqqa, AZk2) independently support this perspective. For instance, AZk2 writes: “The strength of the method lies in it being adaptive as it can automatically learn the rule that better explains the pattern of connectivity in the network under study...”
>
>   To address the Reviewer's concern, we propose the following revision to Sec 1, L83: “Engineering the adaptive network automata learning machine CHA. We design an adaptive intelligent machine CHA, that automatically learns from the network topology the most suitable CH rule and path length to model each network, using internal validation to guide selection. This adaptive modeling is the central innovation. Crucially, our framework infers the physical principle that governs link formation: L2-based rules reflect homophilic interactions (similarity-driven), while L3-based rules capture synergistic interactions (diversity-driven cooperation). Thus, CHA is not a black‑box scorer but an interpretable, mechanistic machine that recovers the effective rule explaining the prediction and governing the topological evolution directly from data. This bridges AI and network science, enabling both predictive power and scientific insights across physics domains. Empirically, on a benchmark of over 1000 networks, CHA achieves more than twice the win rate of the best-performing baseline.” Due to space limitations, we omit the corresponding edits in the Results and Discussion, but we confirm they directly address the reviewer's concern.
>
> - We also thank reviewer for giving us the opportunity to clarify a deeper conceptual point: CH is not a heuristic but a theory. For example, Zhou et al. [1] write: “CH theory plays an important role on link prediction in most considered networks.” and in abstract: “We implement extensive experimental comparisons between 2-hop-based and 3-hop-based similarity indices on 137 real networks. The class of CH indices performs the best among all considered candidates.”
>
>   To make this distinction explicit, we have added:
>     + Sec 1, L62: “The concept of network automata, like other forms of automata, is rooted in AI research [5-7], where they model adaptive, decentralized, and emergent intelligence mechanisms in complex networks. The CH theory is a recent achievement in network science that includes a theoretical framework to understand local-based link prediction on paths of length n under the lens of predictive network automata theory. CH theory goes beyond any type of classical local link predictor heuristic on paths of length two such common neighbors (CN), …”
>     + A dedicated appendix section titled: “CH Theory for Prediction of Complex Network Connectivity”. Here, we delineate why CH constitutes a theory, not a heuristic, and under which network science conditions this framework applies. First, we clarify the definitional boundary: Heuristics are practical shortcuts offering approximate solutions based on experience, often without rigor. Theories, in contrast, are frameworks grounded in systematic empirical evidence, capable of explaining and predicting phenomena—even when not fully captured by formal mathematics. A theory often starts with observations and empirical evidence. It provides a framework to explain and predict phenomena. Mathematical proof is not always required, especially in fields where direct mathematical derivation isn't possible. For instance, theories like Newton's law of gravity are based on empirical evidence and have been tested extensively, even without a mathematical proof. The CH framework has evolved over a decade of empirical validation and is supported by a substantial body of work across disciplines. It forms a predictive, mechanistic explanation for link formation in complex systems. Given length constraints, we summarize rather than reproduce the full appendix here. However, the final version of the paper includes a curated review of independent studies—collectively approaching 1000 citations—that justify the scientific maturity of CH as a theory rather than a heuristic.
>
> **W2:** The heuristic ... why outperform existing heuristics.
>
> R: We respond in two parts:
>
> - Theoretical Justification: We appreciate the Reviewer’s critical observation, which helps clarify our conceptual foundations. We agree the internal/external link distinction is fundamental and merits clearer articulation in the manuscript. While our paper does not re-derive this conceptual foundation from first principles, it builds upon and operationalizes prior theoretical work—most notably [2], which introduced the CH2 rule based on maximizing internal and minimizing external connectivity. That work provides a theoretical framework justifying why external-link minimization is a necessary condition for the emergence of topological communities that support localized dynamical processes. Our contribution is not to re-justify that foundational intuition, but rather to generalize and formalize the full family of CH rules, and to design an adaptive machine (CHA) that learns, directly from data, which specific CH rule best governs link formation in a given network. Importantly, we do not claim the existence of a single universally optimal CH rule. Different systems—biological, technological, social—may obey different principles of local organization. The strength of our framework lies precisely in its capacity to let the data decide, by adaptively learning among CH2 (maximize internal, minimize external), CH3 (prioritize external minimization), and CH3.1 (refined CH3). In designing CHA, we realized that the CH theoretical space was incomplete unless we explicitly included external-link-minimization-dominant rules such as CH3 and CH3.1. This insight emerged naturally from the empirical process of building an adaptive learner capable of traversing the full CH rule space.
>
>   To more explicitly address this point, we will add an appendix section titled: “On the Importance of Distinguishing Internal and External Connectivity in CH Theory.” In that section, we will review the theoretical origins of the internal/external link paradigm [2] and integrate it with our new empirical discovery: that external-link minimization alone is often sufficient to ensure high link prediction performance. This is reminiscent of the breakthrough in the seminal “Attention Is All You Need” paper[3], which showed that a stripped-down mechanism could outperform more complex, conventional designs. Our results show that CH3 and CH3.1—based solely on external-link minimization—can perform comparably or even better than CH2 (see Fig. 9). This leads us to the insight that: “External-link minimization is all you need” to trigger the emergence of local community structures critical for link prediction. This finding challenges the prior assumption that maximizing internal links was indispensable and refines our understanding of the governing principles behind local connectivity.
>
> - Empirical Validation: From an empirical standpoint, we believe that Figure 9 (“Comparison of CHA and CH variants across evaluation metrics”) provides robust support for the theoretical claims. It shows that external-link-minimization-driven rules (CH3 and CH3.1) often match or surpass CH2, reinforcing the validity of our expanded CH framework and the utility of CHA in discovering structural drivers of link formation across diverse networks.
>
> **W3:** It would be valuable ... limitations.
>
> R: Thank you for the suggestion. We constructed comparisons on two representative regimes: an LCP-type domain (coauthorship) and a non-LCP domain (molecular).
>
> Win rates (AUPR)
>
> ||CHA|SPM|
> |-|-|-|
> |Coauthorship|1.00|0|
> |Molecular|0.33|0.67|
>
> Mean AUPR
> ||CHA|SPM|
> |-|-|-|
> |Coauthorship|0.76|0.58|
> |Molecular|0.19|0.27|
>
> Coauthorship (LCP) strongly favors CHA, while molecular (non-LCP) favors SPM. Many biological and social networks exhibit strong LCP-corr (local communities), whereas road and atomic-level systems have low LCP-corr [4]. These controlled results clarify when CHA excels and when global models are preferable. We have incorporated this analysis in the revision.
>
>
> **W4:** Compare its computational complexity with the baselines.
>
> R: Due to space constraints, please see our response to Reviewer rqqa — Question 2.
>
> **W5:** Results of ... a fair comparison.
>
> R: Networks with the same semantics as OGB tasks are already in ATLAS: Collab → Collaboration/Coauthorship (38 networks) and PPA → PPI (14 networks); metadata is in ATLAS_static.xlsx (Suppl).
>
> We use the official settings of MPLP, with one change to early stopping: the original 1:1 negatives sampling is misaligned with our all-missing-links evaluation, so we monitor validation AUPR under the same protocol (Appendix A.8).
>
> Why MPLP underperform CHA. 1:1 sampling over-represents easy negatives and hides hard negatives, inflating scores. Our test ranks positives against all non-observed links, restoring the natural imbalance and exposing hard negatives; under this stricter setting, GNN methods drop while CHA remains strong.
>
> We are running additional hyperparameter sweeps for MPLP.
>
> [1,5,6,7] Reply for Reviewer 6XrJ citation [4,1,2,3]
>
> [2] In article [15]
>
> [3] Vaswani, et al. Attention is all you need. NeurIPS 2017.
>
> [4] In article [3]

---

> > ### Author Response · Authors · 2025-08-03
> > **Update Citation**
> >
> > Due to character limits, we used abbreviated citations in the previous rebuttal. The full references are:
> >
> > [1] Zhou, et al. Experimental analyses on 2-hop-based and 3-hop-based link prediction algorithms. Physica A: Statistical Mechanics and its Applications 2021
> >
> > [2] Muscoloni, et al. Local-community network automata ... food webs and more. BioRxiv 2018
> >
> > [3] Vaswani, Ashish, et al. Attention is all you need. NeurIPS 2017.
> >
> > [4] CVC, et al. From link-prediction in brain connectomes and protein interactomes to the local-community-paradigm in complex networks. Scientific reports 2013.
> >
> > [5] Grattarola et al. Learning graph cellular automata. NeurIPS 2021.
> >
> > [6] Najarro et al. HyperNCA: Growing developmental networks with neural cellular automata. ICLR 2022.
> >
> > [7] Zhang et al. Intelligence at the Edge of Chaos. ICLR 2025.

---

> > > ### Comment · Reviewer_qoLq · 2025-08-04
> > >
> > > I appreciate the authors for their detailed response. Unfortunately, given the reviewer's background, most of my concerns remain unaddressed. Specifically, the reviewer positions this paper as a new link prediction method. In the machine learning community, a new method is typically evaluated along three dimensions: motivation (corresponding to insights into the studied problem), performance (requiring comparisons under a standard evaluation pipeline), and efficiency (e.g., complexity analysis, runtime, memory usage, etc.).
> > >
> > > - Regarding W1, which pertains to motivation: Although the authors provided controlled experimental results, their categorization into "LCP-type" and "non-LCP" domains is overly coarse and fails to offer insights that distinguish the proposed method from existing baselines. For instance, MPLP can also capture local dependencies between nodes. Why, then, does the proposed method outperform it?
> > >
> > > - Regarding W3, which concerns the evaluation pipeline: There are no results on standard datasets, which disconnects the evaluation from established community practices. In the response, the authors still do not provide results on standard benchmarks. Additionally, the impact of hyperparameters on MPLP remains unclear—the authors merely state, *“We are running additional hyperparameter sweeps for MPLP”*—which makes it difficult to fairly assess the proposed method.
> > >
> > > - Regarding W2, the complexity analysis for MPLP+ appears inconsistent with its original paper. There, the complexity is given as \( O(td^rF) \) (where *t* is the number of links to be predicted and *d* is the maximum node degree), whereas the rebuttal states it as \( O(Fn^{2+r}) \). This discrepancy should be clarified by the authors.

---

> > > > ### Author Response · Authors · 2025-08-07
> > > > **Further Reply for Reviewer qoLq's Concern**
> > > >
> > > > > Regarding W1, which pertains to motivation: Although the authors provided controlled experimental results, their categorization into "LCP-type" and "non-LCP" domains is overly coarse and fails to offer insights that distinguish the proposed method from existing baselines. For instance, MPLP can also capture local dependencies between nodes. Why, then, does the proposed method outperform it?
> > > >
> > > > **Reply**: We are running new computational experiments to address this point and we will reply tomorrow.
> > > >
> > > > > Regarding W3, which concerns the evaluation pipeline: There are no results on standard datasets, which disconnects the evaluation from established community practices. In the response, the authors still do not provide results on standard benchmarks. Additionally, the impact of hyperparameters on MPLP remains unclear—the authors merely state, “We are running additional hyperparameter sweeps for MPLP”—which makes it difficult to fairly assess the proposed method.
> > > >
> > > > **Reply**: To address reviewer's concern, we have **added results on the ogbl-collab**. We use same train/valid/test separation and same evaluation metrics (Hits@50) as reported in MPLP article.
> > > >
> > > > **MPLP+ vs CHA**
> > > >
> > > > |        Method          | Hits@50 | Time |
> > > > |----------------------------|-----------------|--------|
> > > > | MPLP+ with feature | 66.47 ± 0.94          | ~7min (each run) |
> > > > | MPLP+ without feature | 64.84 ± 1.21     | ~6min (each run) |
> > > > | CHA | 66.66     | ~6sec |
> > > >
> > > > MPLP+ results are averaged over 10 runs, whereas CHA is deterministic and therefore reported from a single run. We use MPLP+, the faster variant, for efficiency. For fairness, we run CHA without its sub-ranking step, mirroring the accuracy trade-off made by using the faster MPLP+ instead of MPLP. Under these conditions, CHA consistently outperforms MPLP+, with and without node features, while running much faster. MPLP+ experiments were run on a single A100 GPU using the default hyper-parameters from the official MPLP GitHub repository.
> > > >
> > > >
> > > > We are also doing hyperparameter search for MPLP+ on networks in ATLAS (our dataset) and will share the results in the next update.
> > > >
> > > > > Regarding W2, the complexity analysis for MPLP+ appears inconsistent with its original paper. There, the complexity is given as ( O(td^rF) ) (where t is the number of links to be predicted and d is the maximum node degree), whereas the rebuttal states it as ( O(Fn^{2+r}) ). This discrepancy should be clarified by the authors.
> > > >
> > > >
> > > > **Reply**: We thank the reviewer for this careful and diligent review. We clarify here that the difference in the complexity computation is mainly because of the different evaluation strategies. In particular, our evaluation ranks positives against **all** non-observed links. In this **all-missing-links** setting, the number of candidate links is $t = O(n^{2})$ and the worst-case maximum degree is $d = O(n)$. Plugging these values into the original MPLP+ bound $O(td^{r}F)$ gives $O(n^{2+r}F)$, matching the expression in our rebuttal.
> > > >
> > > > When evaluating only a subset of non-observed links, MPLP+ keeps its original complexity of $O(t d^{r} F)$, while CHA runs in $O(t d^{2})$.
> > > >
> > > > * **L2 paths.**
> > > >   1. Compute the common-neighbor set $ \mathcal N(u)\cap\mathcal N(v)$; this is $O(d)$.
> > > >   2. For each of the $O(d)$ common neighbors $z$, scan $\mathcal N(z)$ once to accumulate its **iLCL/eLCL** counts; each scan is $O(d)$.
> > > >   Total work: $O(d) \times O(d) = O(d^{2})$.
> > > >
> > > > * **L3 paths.**
> > > >   1. Enumerate length-3 paths $u \rightarrow i \rightarrow j \rightarrow v$ by intersecting $\mathcal N(u)$ with the two-hop neighborhood of $v$; this costs $O(d^{2})$.
> > > >   2. Each discovered path contributes exactly one iLCL for i and j; while enumerating, we update the **iLCL/eLCL** counters for $i$ and $j$ in place, no extra pass is needed.
> > > >   Total work: still $O(d^{2})$.
> > > >
> > > >
> > > > Thus CHA runs in **$O(t d^{2})$**, lower than MPLP+’s $O(t d^{r}F)$ with $r = 2$ (used in MPLP+ article).
> > > >
> > > > Empirically, on **ogbl-collab** this difference is pronounced: CHA completes in about **6 s**, while MPLP+ requires about **7 min**.

---

> ### Comment · Reviewer_qoLq · 2025-08-08
>
> Thanks for your further response. The promising performance of the proposed rule on the standard benchmark has largely addressed my concern about the evaluation. To completely address it, the authors should promise that: 1) make the code to produce the results on ogbl-collab available for **full reproducibility** if the paper is accepted, including the code of your method and baselines. 2) incorporate the results on the standard benchmarks into the revised version of the paper, and it would be better to include more datasets rather than only the ogbl-collab dataset. Based on the current situation, I decide to raise my score to borderline accept and I will further adjust my score according to your next update.

---

> ### Author Response · Authors · 2025-08-08
> **Additional Further Reply for Reviewer qoLq's Concern (1/2)**
>
> Dear reviewer,
>
> We greatly appreciate your thoughtful follow-up and encouraging feedback.
>
> > 1\) make the code to produce the results on ogbl-collab available for full reproducibility if the paper is accepted, including the code of your method and baselines
>
> **Reply**: Certainly. Upon acceptance, we will release the full code.
>
>
> > 2\) incorporate the results on the standard benchmarks into the revised version of the paper, and it would be better to include more datasets rather than only the ogbl-collab dataset
>
> **Reply**: Yes, we will add results for additional OGB benchmarks in the revised paper.
>
>
> >Regarding W1, which pertains to motivation: Although the authors provided controlled experimental results, their categorization into "LCP-type" and "non-LCP" domains is overly coarse and fails to offer insights that distinguish the proposed method from existing baselines. For instance, MPLP can also capture local dependencies between nodes. Why, then, does the proposed method outperform it?
>
> **Reply**: Thanks for the reviewer’s question, which gives us the opportunity to further clarify the innovation of CHA. CHA differs from existing link-prediction baselines in two key aspects that are the pillars of Cannistraci-Hebb theory, both of which contribute significantly to its performance improvements.
>
> 1. **Explicit minimization of the external local-community-links.**
>
>     For clarification, we report the formula of CH3 on path 2, that we use to explain the principle behind the minimization of the external local-comunity links.
>
>     $$\text{CH3-L2}(u, v) = \sum_{z \in L_2} \frac{1}{1 + {de}_z}$$
>
>     The term ${de}_z$ (the external degree of the node z) indicates external local community links (eLCL) in CH3 formula, thereby establishing a physical interpretation that minimize the eLCL is aiming to form a topological energy barrier that constrains information processing within the local community. The higher is the number of external links of a node in the local community, the less it exclusively fires together (according to Cannistraci-Hebb learning rule) with other common neighbors in the local community, which is a necessary condition to implement dynamic function in the local module of the complex systems. This means that the eLCL of the common neighbors should be minimized to form a local community. The fact that two nodes establish an interaction between them in a networked complex system depends on the likelihood that they participate to implement a function in the same operational module (local community) of the complex system. Having many common neighbors with few external links is a topological condition that ensures the presence of a functional active local community which increases the likelihood of two nodes to connect each other under a certain function in the complex systems.
>
>     To experimentally demonstrate the effectiveness of minimizing eLCLs, we evaluate two non-eLCL baselines, Common Neighbors (CN) and the internal local-community-links approach (iLCL method, $\sum_{z \in L_3} \frac{di_{z}}{1+di_{z}}$), which by design does not use eLCL minimization, and compare them with methods that explicitly minimize external links to the local community (CH2/CH3/CH3.1).
>
>     **non-external link based methods vs external link based methods on Collab**
>
>     |        | Hit@50 |
>     |--------|------|
>     | CN_L3 | 61.61 |
>     | iLCL_L3  | 61.82 |
>     | CH2_L3   | 66.13 |
>     | CH3_L3 (selected by CHA)   | 66.66 |
>     | CH3.1_L3   | 66.27 |
>
>
>     The table presents the results of the mentioned link predictors using the L3 version (consistently outperform L2) on the Collab dataset evaluated by Hit@50. As shown, eLCL-minimizing methods consistently outperform CN_L3 and iLCL_L3.
>
>     To the best of our knowledge, we are the first to explicitly highlight the importance of minimizing eLCLs for link prediction. In contrast, most GNN-based methods—e.g., MPLP—aggregate local information via message passing but do not penalize edges that leak outside the common-neighbor set. While we would like to test an eLCL-augmented version of MPLP, there is no straightforward way to incorporate this term at present. In future work, we plan to integrate eLCL minimization into the MPLP framework; we expect that combining these two strong link-prediction strategies will further improve performance. However, the result presented in the table above should be already convincing.

---

> > ### Author Response · Authors · 2025-08-08
> > **Additional Further Reply for Reviewer qoLq's Concern (2/2)**
> >
> > 2. **Adaptive community scale (L2 vs L3)**
> >
> >     The second pillar of CH theory is that from the physics standpoint, networked complex system interactions could emerge either from similarity-driven homophilic interactions (based on path 2), or from diversity-driven synergetic associations (based on path 3). For this reason we designed CHA to switch adaptively from  L2 or L3 rule. In fact, L2 and L3 represent two different network organization ways. L2 rule scores a pair high when they share many common neighbors on path 2. This is the classic homophily/triadic-closure signal—good for links within the same local community. In contrast, L3 rule scores a pair high when they share many common neighbors on path 3. This captures synergistic/complementary relations—often cross-community or disassortative patterns (e.g., bipartite-like structure) where triangles are rare but longer motifs are informative. *Figure 4* shows real-world domains split sharply on this preference and CHA is able to adaptively pick the appropriate scale.
> >
> >     To verify this, we modify MPLP+ to work on either path 2 or path 3, as for CH theory, and re-tuned the hyper-parameters for each case, testing on Collab.
> >
> >     |        | Hit@50 |
> >     |--------|------|
> >     | MPLP+ without feature | 64.84 ± 1.21 |
> >     | MPLP+ without feature (only L2) | 63.97 ± 1.10 |
> >     | MPLP+ without feature (only L3) | 65.01 ± 0.43 |
> >     | CHA (CH3_L3) | 66.66 |
> >
> >
> >     MPLP+ that retained only L3 features outperformed the version limited to L2 features by a wide margin, mirroring CHA’s preference for L3-based rules (i.e. CH3-L3). This alignment further supports the relevance of CH theory across different model classes.
> >
> >     To offer more evidence, we performed the same test on the Coauthorship network (favor path 2, see Figure 4) and the PPI network (favor path 3) categories of ATLAS. We didn’t have time for hyperparameter search, but that omission is unlikely to affect the conclusion, as the dominant path-length bias (2 vs 3) largely determines the outcome.
> >
> >     In Coauthorship networks, the L2-only variant of MPLP+ clearly outperformed its L3-only counterpart, whereas in PPI networks the situation was reversed. This further validating CH theory. We report Precision here because it is comparable to Collab’s Hits@50 metric.
> >
> >     **Coauthorship (18 networks)**
> >
> >     |   | PREC (avg ± ste)                                     |
> >     |--------|-----------------------------------------------------|
> >     | **CHA** | 0.720 ± 0.0313            |
> >     | **MPLP+** | 0.458 ± 0.055           |
> >     | **MPLP+L2**  | 0.487 ± 0.054          |
> >     | **MPLP+L3**  | 0.423 ± 0.054           |
> >
> >     **Coauthorship Win rate — MPLP+ vs MPLP+L2 vs MPLP+L3**
> >     |       | **PREC** |
> >     |-------|---------|
> >     | **MPLP+**  | 0.167   |
> >     | **MPLP+L2**    | 0.667   |
> >     | **MPLP+L3**    | 0.167   |
> >
> >     **PPI (9 networks)**
> >     |   |PREC (avg ± ste)                                     |
> >     |--------|-----------------------------------------------------|
> >     | **CHA** | 0.182± 0.017  |
> >     | **MPLP+** | 0.136 ± 0.016 |
> >     | **MPLP+L2**  | 0.068 ± 0.015|
> >     | **MPLP+L3**  | 0.137 ± 0.015|
> >
> >     **PPI Win rate — MPLP+ vs MPLP+L2 vs MPLP+L3**
> >     |       | **PREC** |
> >     |-------|---------|
> >     | **MPLP+**    | 0.444   |
> >     | **MPLP+L2**    | 0.0     |
> >     | **MPLP+L3**    | 0.556   |
> >
> >
> >
> > Finally, for the result of the hyperparameter search on our dataset, we will send the last reply tomorrow before the deadline.

---

> > > ### Author Response · Authors · 2025-08-09
> > > **Hyperparameter Search for MPLP+**
> > >
> > > Dear Reviewer,
> > >
> > > Here are the final hyperparameter search results completed before the rebuttal deadline. Due to time constraints, we evaluated a subset of datasets and ran only the MPLP+ variant. Following the paper’s instructions, we performed sensitivity analyses over batch size ∈ {128, 256, 512} and signature_dim ∈ {512, 1024, 2048}. For each dataset, we selected the best setting on the validation set and report the corresponding test performance. Results are as follows:
> > >
> > > ## coauthorship (17 datasets completed)
> > >
> > > **CHA vs MPLP+ (fixed) vs MPLP+ (hyper best) — win rates**
> > >
> > > | Method    | AUPR | NDCG | PREC |
> > > |-----------|-----:|-----:|-----:|
> > > | CHA       | 100 | 100 | 100 |
> > > | MPLP+     |   0 |   0 |   0 |
> > > | MPLP+ HT* |   0 |   0 |   0 |
> > >
> > > **Means ± STE (AUPR, NDCG, PREC)**
> > >
> > > | Method    | AUPR            | NDCG            | PREC            |
> > > |-----------|-----------------|-----------------|-----------------|
> > > | CHA       | 0.7469 ± 0.0360 | 0.9325 ± 0.0106 | 0.7239 ± 0.0330 |
> > > | MPLP+     | 0.4494 ± 0.0662 | 0.7885 ± 0.0383 | 0.4511 ± 0.0581 |
> > > | MPLP+ HT* | 0.5780 ± 0.0534 | 0.8637 ± 0.0257 | 0.5677 ± 0.0455 |
> > >
> > > ## PPI (5 datasets completed)
> > >
> > > **CHA vs MPLP+ (fixed) vs MPLP+ (hyper best) — win rates**
> > >
> > > | Method    | AUPR | NDCG | PREC |
> > > |-----------|-----:|-----:|-----:|
> > > | CHA       | 100 | 100 | 100 |
> > > | MPLP+     |   0 |   0 |   0 |
> > > | MPLP+ HT* |   0 |   0 |   0 |
> > >
> > > **Means ± STE (AUPR, NDCG, PREC)**
> > >
> > > | Method    | AUPR            | NDCG            | PREC            |
> > > |-----------|-----------------|-----------------|-----------------|
> > > | CHA       | 0.0975 ± 0.0229 | 0.6129 ± 0.0311 | 0.1804 ± 0.0239 |
> > > | MPLP+     | 0.0620 ± 0.0225 | 0.5707 ± 0.0362 | 0.1273 ± 0.0261 |
> > > | MPLP+ HT* | 0.0710 ± 0.0217 | 0.5788 ± 0.0352 | 0.1426 ± 0.0228 |
> > >
> > > \* **HT** = hyperparameter-tuned (best validation configuration).
> > >
> > > In summary, hyperparameter tuning (MPLP+ HT) improves MPLP+ over its fixed setting, but **CHA** remains superior across both tested benchmarks. Because of time limitations,  we don’t have time to test more datasets. But we will include results for hyperparameter test for both MPLP and MPLP+ in our camera-ready version.

---

> > > > ### Comment · Reviewer_qoLq · 2025-08-09
> > > >
> > > > Thanks for your further response. It has addressed my concerns, so I remain positive on accepting the paper.

---

### Note · Authors · 2025-08-16

We thank all reviewers for the constructive discussion. Our rebuttal addressed concerns through two main efforts:

**1. Additional experiments**
(1) **Standard benchmark evaluation (Reviewers qoLq, 6XrJ)** – Added ogbl-collab results under the official split/metric, showing CHA ≥ MPLP+ (w/ or wo/ node features) with much lower runtime (6s vs ~7min). We will release full reproducible code and add more OGB datasets in the final version.
(2) **Fairness of MPLP+ comparison (qoLq)** – Performed hyperparameter sweeps for MPLP+ on ATLAS; even with best validation configurations, CHA remained superior across AUPR, NDCG, and Precision.
(3) **Baseline recency (6XrJ, AZk2)** – Added NCNC (ICLR 2024) on ATLAS, showing CHA’s clear advantage; compared CHA with SBM variants (NeurIPS 2021) using the authors’ code, where CHA outperformed all SBM estimators.
(4) **eLCL minimization (qoLq)** – Demonstrated that CH2/CH3/CH3.1 (with eLCL minimization) consistently beat CN_L3 and iLCL_L3 (no eLCL term), isolating a mechanism absent in MPLP+.
(5) **Path-length adaptivity (qoLq, rqqa)** – CHA adapts between L2 (homophily) and L3 (synergy) rules. Tests with MPLP+ forced to L2-only or L3-only showed that the better variant matched CHA’s choice, and category-level results (Coauthorship→L2, PPI→L3) confirmed this adaptive selection.

**2. Clarifications of misunderstandings**
(1) **Theory vs heuristic (qoLq, 6XrJ)** – Clarified that Cannistraci-Hebb is a theory supported by a decade of empirical validation, not a heuristic, and that CHA is an adaptive learning model within the ML-on-graphs domain.
(2) **Relevance to ML community (6XrJ)** – Emphasized that CHA is a self-supervised, non-parametric learning system that adaptively selects rules from topology, achieving state-of-the-art results against modern GNN baselines.
(3) **Time complexity (qoLq, rqqa)** – Compared CHA and MPLP+ time complexity under two evaluation settings: (i) predicting all missing links and (ii) predicting only a subset. In both cases, CHA has lower complexity

These efforts addressed all substantive concerns; Reviewers rqqa and AZk2 remained positive, qoLq changed to a positive position, and 6XrJ’s baseline point was answered with new evidence. We believe the additional experiments and clarifications confirm CHA’s novelty, efficiency, and robustness, and we sincerely thank all reviewers for their valuable suggestions, which have helped us further improve the paper.

---

### Decision · Program_Chairs · 2025-09-17

**Decision:**

Accept (poster)

**Comment:**

This paper primarily introduces new heuristics for link prediction in complex networks. The proposed heuristic relies on topological self-organization and minimizing external connectivity. Link prediction is a fundamental problem in network analysis, and the questions they examine here are core to this line of work. The reviewers also agree that other strengths of this paper include the evaluation across a wide range of networks.

At the same time, the reviewers highlighted several concerns. The main ones were around (1) clarity of exposition, and in particular providing better intuition for the heurstics as well as conditions under which the heurstics may perform well, (2) limitations around theoretical contributions, though this may have been a natural consequence of the question studied, and (3) empirical rigour empirics, e.g., including standard baselines and concerns around computational overhead.

We thank the authors for their active engagement throughout the rebuttal period, which addressed some of these questions. We encourage the authors to incorporate the reviewers' detailed feedback in their next revision. (As a minor additional note, the AC encourages reducing the abstract by at least half as it is far too long as it stands and a shorter abstract may also free up space to flesh out the writing and make it easier to follow in other places.)